# Exceptional floods in the Prut basin, Romania, in the context of heavy rains in the summer of 2010

Gheorghe Romanescu[1], Cristian Constantin Stoleriu

Alexandru Ioan Cuza, University of Iasi, Faculty of Geography and Geology, Department of Geography, Bd. Carol I, 20 A, 700505 Iasi, Romania

**Abstract.** The year 2010 was characterized by devastating flooding in Central and Eastern Europe, including Romania, the Czech Republic, Slovakia, and Bosnia-Herzegovina. This study focuses on floods that occurred during the summer of 2010 in the Prut River basin, which has a high percentage of hydrotechnical infrastructure. Strong floods occurred in eastern Romania on the Prut River, which borders the Republic of Moldova and Ukraine, and the Siret River. Atmospheric instability from 21 June-1 July 2010 caused remarkable amounts of rain, with rates of 51.2 mm/50 min and 42.0 mm/30 min. In the middle Prut basin, there are numerous ponds that help mitigate floods as well as provide water for animals, irrigation, and so forth. The peak discharge of the Prut River during the summer of 2010 was 2,310 m$^3$/s at the Radauti Prut gauging station. High discharges were also recorded on downstream tributaries, including the Baseu, Jijia, and Miletin. High discharges downstream occurred because of water from the middle basin and the backwater from the Danube (a historic discharge of 16,300 m$^3$/s). The floods that occurred in the Prut basin in the summer of 2010 could not be controlled completely because the discharges far exceeded foreseen values.

## 1 Introduction

Catastrophic floods occurred during the summer of 2010 in Central and Eastern Europe. Strong flooding usually occurs at the end of spring and the beginning of summer. Among the most heavily affected countries were Poland, Romania, the Czech Republic, Austria, Germania, Slovakia, Hungary, Ukraine, Serbia, Slovenia, Croatia, Bosnia and Herzegovina, and Montenegro (Bissolli et al., 2011; Szalinska et al., 2014) (Fig. 1). The strongest floods from 2010 were registered in the Danube basin (see Table 1). For Romania, we underlined the floods from the basins of Prut, Siret, Moldova and Bistrita rivers. The most devastating floods in Romania occurred in Moldavia (Prut, Siret) and Transylvania (Tisa, Somes, Tarnave, Olt). The most deaths were recorded in Poland (25), Romania (six on the Buhai River, a tributary of the Jijia), Slovakia (three), Serbia (two), Hungary (two), and the Czech Republic (two) (Romanescu and Stoleriu, 2013a,b).

Floods are one of the most important natural hazards in Europe (Thieken et al., 2016) and on earth as well (Merz et al., 2010; Riegger et al., 2009). They generate major losses in human lives, and also property damage (Wijkman and Timberlake, 1984). For this reason, they have been subject to intense research, and significant funds have been allocated to mitigating or stopping them. According to Merz et al. (2010) "the European Flood Directive on the assessment and management of flood risks (European Commission, 2007) requires developing management plans for areas with significant flood risk (at a river basin scale), focusing on the reduction of the probability of flooding and on the potential consequences to human health, the environment and economic activity." (p. 511). Several studies investigated catastrophic floods or the floods that generated significant damage. They focused on: the statistical distribution of the maximum annual discharge, using GEV and the links with the

---

[1] Corresponding author: romanescugheorghe@gmail.com

basin geology (Ahilan et al., 2012); climate change impacts on floods (Alfieri et al., 2015;
Detrembleurs et al., 2015; Schneider et al., 2013; Whitfield, 2012); disastruous effects on
infrastructures such as transportation infrastructures, and their interdependence (Berariu et al.,
2015); historical floods (Blöschl et al., 2013; Strupczewski et al., 2014; Vasileski and
Radevski, 2014) and their links to heavy rainfall (Bostan et al., 2009; Diakakis, 2011;
Prudhomme and Genevier, 2011; Retsö, 2015); the public perception of flood risks (Brilly and
Polic, 2005; Feldman et al., 2016; Rufat et al., 2015); land use changes and flooding
(Cammerer et al., 2012); the evolution of natural risks (Hufschmidt et al., 2005);
geomorphological effects of floods in riverbeds (Lichter and Klein, 2011; Lóczy and
Gyenizse, 2011; Lóczy et al., 2009, 2014; Reza Ghanbarpour et al., 2014); the spatial
distribution of floods (Moel et al., 2009; Parker and Fordham, 1996); the interrelation
between snow and flooding (Revuelto et al., 2013).

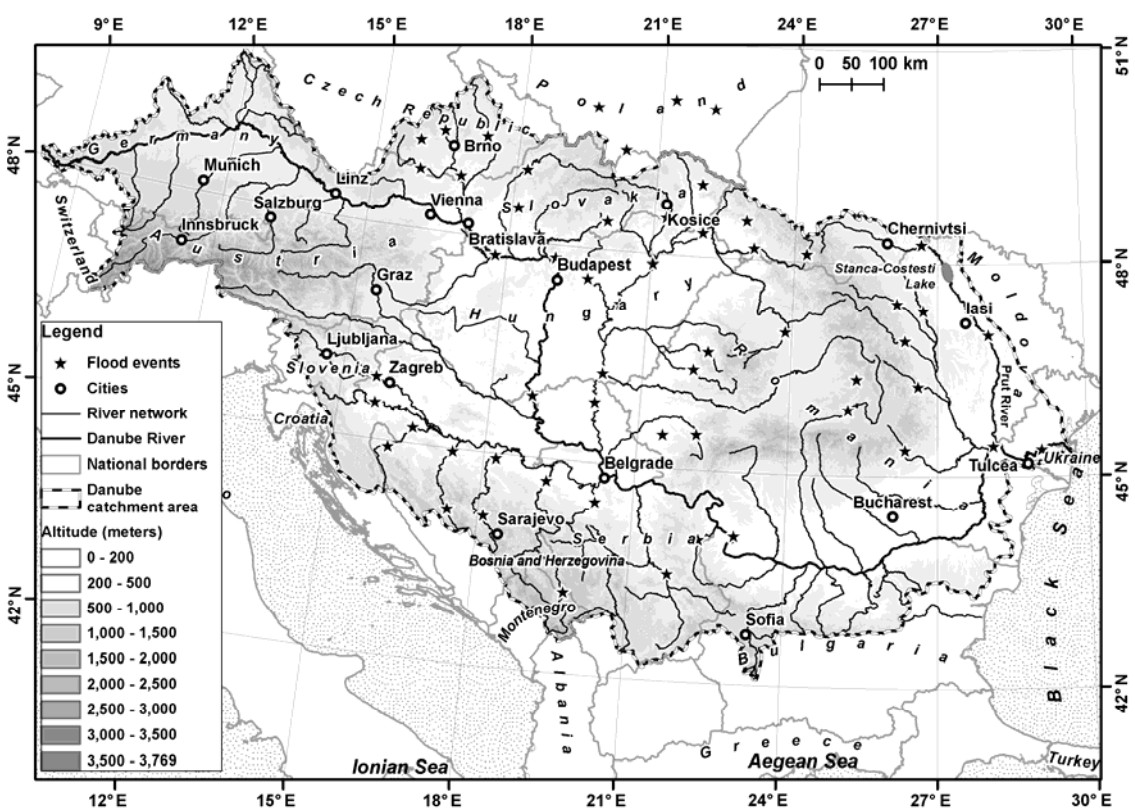

**Figure 1.** The Danube catchment and the location of the most important floods that occurred
from May-June 2010
**Table 1.** Overview of main flood events for the Danube river basin in 2010, as forecasted by
EFAS and/or reported in international on-line news media (ICPDR, 2010)

| From (dd.mm) | To (dd.mm) | River Basin Afected | Country Affected | EFAS Alert Sent? | Date FAS Alert Sent | Confirmed? | Comment |
|---|---|---|---|---|---|---|---|
| 20.II | 4.III | Sava | HR/ RS | Yes (Flood Watch) | 24 Feb. | Yes | Severe flooding in Central & E. Serbia, and in Sava & Morava river systems. |
| 21.II | 28.II | Velika | RS | Yes | 16 Feb. | Yes | Severe flooding in eastern |

| | | | | (Flood Watch) | | | Serbia |
|---|---|---|---|---|---|---|---|
| | | Morava | | | | | |
| Febr. | Febr. | Koeroes | RO/HU | Yes (Flood Watch) | 16 Feb. | No | (No reports found on on-line news media). Events to be confirmed by partners in next annual EFAS meeting |
| 1.III | 5.III | Danube | RO/BG | Yes (Flood Alert) | 3 Mar. | Yes | Severe flooding in S. Romania and in N.W. & N. Bulgaria. |
| March | March | Somes/Mures/Koeroes | RO/HU | Yes (Flood Alert) | 18 Mar. | No | No reports found on on-line news media. Events to be confirmed by partners in next annual EFAS meeting |
| 15.V | 30.V | Danube/Oder | SK/PL/CZ/HU | Yes (Flood Alert) | 12 May. | Yes | Extensive flooding in central & eastern Europe, esp. Poland, Czech Republic, Slovakia, Hungary and Serbia. |
| Late June | July | Siret/Prut/Moldova/Bistrita | RO/MD | No | - | Yes | Severe flooding in N.E. Romania kill 25 people, also some counties in Moldova. |
| 15.VII | 15.VII | Prut/ Olt | RO | Yes (Flood Alert) | 7 July. | Yes | Maximum flood alert on Prut river in E. Romania, along border with Moldova. |
| 17.IX | 19.IX | Sava/Soca | HR/SL | Yes (Flood Alert) | 18 Sept. | Yes | Severe flooding in Slovenia kill 3 people. Croatia also affected. |
| Late Nov. | Early Dec. | Drina | RS | Yes (Flood Alert) | 29 Nov. | Yes | Severe flooding in Bosnia, Serbia and Montenegro, with river Drina at highest level in 100 years. |
| 3.XII | 8.XII | Sava | HR | Yes (Flood Alert) | 5 Dec. | Yes | Heavy rain causes devastating flooding in the Balkans, esp. Bosnia and Herzegovina, Croatia, Montenegro, & Serbia. |
| 9.XII | 9.XII | Tisza | HU/RS | No | - | Yes | Snow-melt and swollen rivers flood 3000 km2 of arable land, esp. near Szeged, on Tisza river, in S.E. Hungary. |
| Dec. | Dec. | Koeroes | HU/RO | Yes (Flood Alert) | 3 Dec. | No | (No reports found on on-line news media. Event to be confirmed by local authorities in annual EFAS meeting) |

The Prut catchment basin spans three topographic levels: mountains, plateaus, and plains. The surface and underground water supply to the Prut varies by region and is extremly influenced by climatic conditions. This study underscores the role played by local heavy rains in the occurrence of floods, as well as the importance of ponds, mainly the Stanca-Costesti reservoir, in the mitigation of backwaters. We also analyse the local contribution of each catchment basin on the right side of the Prut to the occurrence of the exceptional floods in the summer of 2010. Finally, we consider the upstream discharge and its influence on the lower reaches of the Prut.

## 2 Study area

The Prut River's catchment is situated in the northeastern Danube basin. It is surrounded by several other catchments: the Tisa to the northeast (which spans Ukraine, Romania, and Hungary), the Siret to the west (which is partially in Ukraine), and the Dniestr (in the Republic of Moldova) to the northeast. The Prut catchment occupies eastern Romania and the western part of the Republic of Moldova (Fig. 2). The Prut River begins in the Carpathian Mountains in Ukraine and empties into the Danube near the city of Galati. The catchment measures 27,500 km$^2$, of which 10,967 km$^2$ lies in Romania (occupying approximately 4.6% of the surface of Romania).

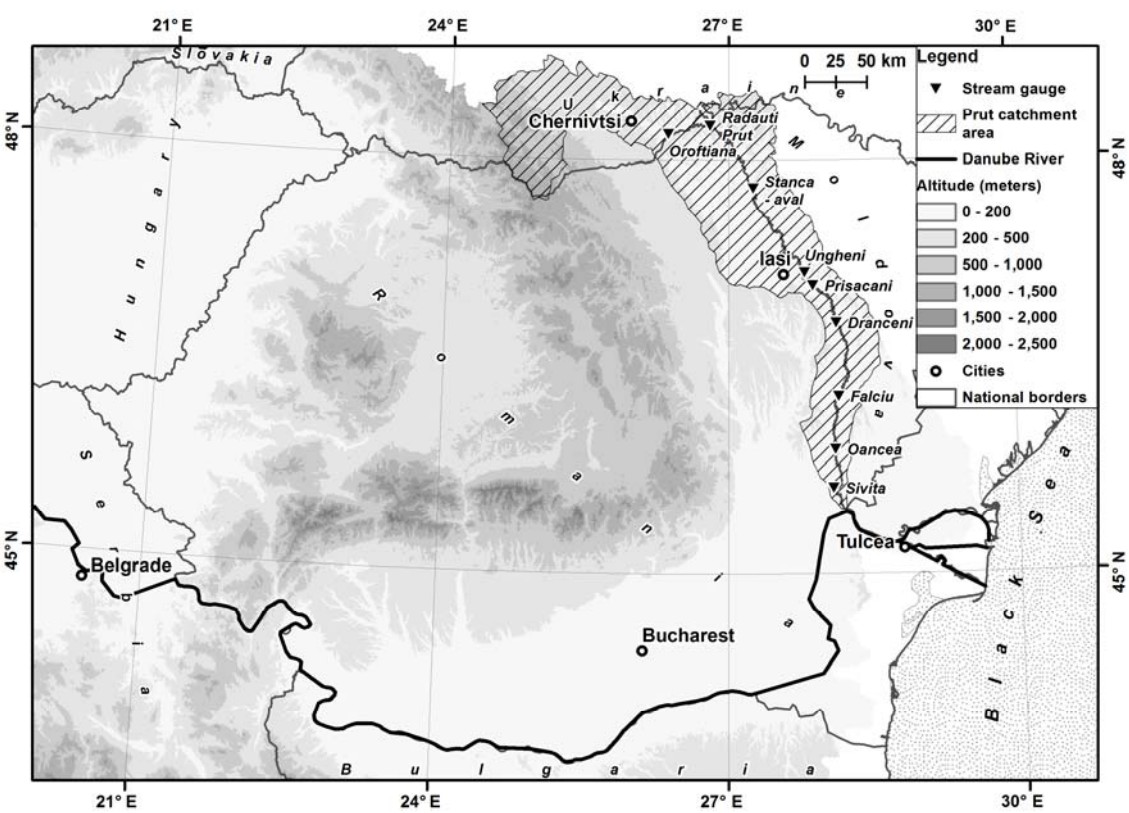

**Figure 2.** Geographic position of the Prut catchment basin in Romania, Ukraine, and the Republic of Moldova, and distribution of the main gauging stations

The Prut River is the second-longest river in Romania, at 952.9 km in length. It is a cross-border river, with 31 km in Ukraine and 711 km in the Republic of Moldova. The mean altitude of the midstream sector of catchment area is 130 m, and for the downstream sector is 2 m. The Prut has 248 tributaries. Its maximum width is 12 km (in the lower reaches, Brates Lake) and its average slope is 0.2%. Its hydrographic network measures 11,000 km in total, of which 3,000 km are permanent streams (33%) and 8,000 km are intermittent (67%). The network has the highest density in Romania at 0.41 km/km$^2$ (the average density is 0.33 km/km$^2$).

The Prut catchment is relatively symmetrical, but its largest proportion is in Romania. To the west, it has 27 tributaries, including the Poiana, Cornesti, Isnovat, Radauti, Volovat, Baseu, Jijia (with a discharge of 10 m$^3$/s, the most important), Mosna, Elan, Oancea, Branesti, and Chineja. The Jijia River is 275 km long, has a catchment area of 5757 km$^2$ and

an annual average flow of 14 m$^3$/s. Its most important tributaries are Miletin, Sitna and
Bahlui. To the east, it has 32 tributaries, including the Telenaia, Larga, Vilia, Lopatnic,
Racovetul, Ciugurlui, Kamenka, Garla Mare, Frasinul, and Mirnova (Romanescu et al.,
2011a,b). The catchment basin has 225 small ponds, counting the Dracsani, which is the
largest pond in Romania. Small ponds are used as drinking water for livestock or to irrigate
subsistence rural households. They usually belong to individual households. Large ponds, on
the other hand, have multiple uses, such as: flooding mitigation, irrigation, fish farming etc.
They resisted better in time because of their significant surface and depth. Large ponds belong
to rural or urban communities. The river also has 26 large ponds, of which the most important
is the Stanca-Costesti reservoir, which has the largest water volume of the interior rivers in
Romania (1,400 million m$^3$).
The topography of the Prut basin includes the Carpathians in the spring area and the
Moldavian Plateau and the Romanian Plain near the river mouth. Arable land occupies 54.7%
of the Prut catchment, while forests occupy 21.4%, perennial cultures occupy another 13.3%,
and the water surface occupies only 1.19%. The mean annual temperature in the Prut
catchment is 9°C, and the mean annual precipitation is 550 mm. The mean annual discharge
increases downstream, varying from 82 m$^3$/s at Radauti Prut to 86.7 m$^3$/s at Ungheni to 93.8
m$^3$/s at the Oancea gauging station situated near the mouth over the period 1950-2008.
Discharges in the downstream reaches of the Prut are controlled by the Stanca-Costesti
reservoir. In the Romanian Register of Large Dams, the Stanca-Costesti dam ranks 49[th] out of
246 dams in terms of height, but 2[nd] in terms of active reservoir volume (1,400 million m$^3$,
after the Iron Gates I, with a volume of 2,100 million m$^3$). It has a surface area of 5,900 ha
during a normal retention level (NRL). After construction of the Stanca-Costesti reservoir,
floods on the Romanian parts of the Prut diminished considerably. Because the Prut has
higher banks in the Republic of Moldova, this area was not affected by dam construction. The
reservoir was constructed with a mitigation level of 550 million.m$^3$, allowing the mitigation of
a 1% backwater from 2,940 to 700 m$^3$/s. The damming infrastructure constructed downstream
from the hydrotechnical nodes prevents the flooding of approximately 100,000 ha of
floodplain area (Romanescu et al., 2011a,b).
**3 Methodology**
Diverse methodology has been used to analyse exceptional floods. Hydrological data,
including discharge and the water level, were obtained from the Prut-Barlad Water Basin
Administration based in Iasi (a branch of the "Romanian Waters" National Administration).
For catchment basins that did not have gauging stations or observation points, measurements
were taken to estimate the discharge. Mathematical methods were used to reconstitute
discharges and terrain measurements using land surveying equipment (Leica Total Station)
were used to calculate the surface of the stream cross-section. Most stations within the
Romanian portion of the Prut catchment are automatic (Fig. 3). The recording and analysing
methodology used is standard or slightly adapted to local conditions: e.g. the influence of
physical-geographical parameters on runoff (Ali et al., 2012; Kappes et al., 2012; Kourgialas
et al., 2012; Waylen and Laporte, 1999); the management of risk situations (Delli-Priscoli and
Stakhiv, 2015; Demeritt et al., 2013; Grobicki et al, 2015 Grobicki et al, 2015); the role of
reservoirs in flood mitigating (Fu et al., 2014; Serban et al., 2004; Sorocovschi, 2011); the
probability of flooding and the changes in the runoff regime (Hall et al., 2004, 2014; Jones,
2011; Seidu et al., 2012a,b; Wu et al., 2011); flood prevention (Hapuarachchi et al., 2011);
runoff and stream flow indices (Nguimalet and Ndjendole, 2008); morphologic changes of
riverbeds or lake basins (Rusnák and Lehotsky, 2014; Touchart et al., 2012; Verdu et al.,
2014) etc.
The cartographic basis used to map altitudes and slopes is Shuttle Radar Topography Mission
(Global Land Cover Facility, 2016), at a 1:50000 scale. The vector layers were projected
within a geodatabase, using ArcGis 10.1. They include stream lines, sub-catchment basins,
and reservoirs and ponds polygons, as well as gauging station points. In order to generate the
GIS layers, we applied the following methods: digitisation, queries, conversion, geometries
calculation (length, surface) and spatial modelling. Water levels and discharges data were
processed and plotted on charts using the Open Office software. We also used the Inkscape
software          to          design          the          final          maps          and
images.

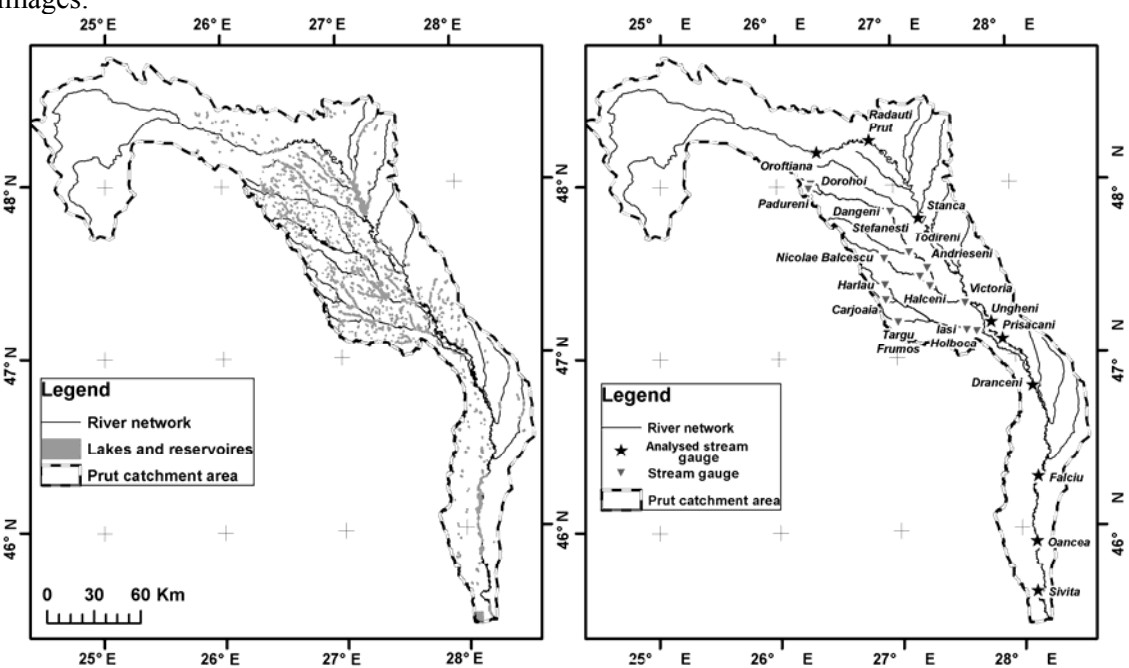

**Figure 3.** Main tributaries, reservoirs (left), and gauging stations (right) in the Prut River
basin

All areas with gauging stations had automatic rain gauges (Anghel et al., 2011;
Tirnovan et al., 2014a,b) (Fig. 3, Table 2). The heavy rains that cause flooding are recorded
hourly over the course of 24 hours according to the Berg intensity scale (Berg et al., 2009). In
the areas lacking gauging stations, data were collected from the closest meteorological
stations, which are automatic and form part of the national monitoring system. The water
level and discharge were analysed throughout the entire flood period. For comparison, the
mean monthly and annual data for the water level and discharge were also analysed. The
processed data were portrayed as histograms that illustrate the evolution of water levels
during the floods, including the CA (warning level), CI (flood level), and CP (danger level)
flood threshold levels before and after the flood, the daily and monthly runoff, and the hourly
variations of runoff during the backwater. For an exact assessment of the damage and the
flooded surface area, observations and field measurements were conducted on the major
floodplains of the Volovat, Baseu, Jijia, Sitna, Miletin, Bahluet, Bahlui, Elan, and Chineja
Rivers (Romanescu and Stoleriu, 2013b).
Nine gauging stations exist in Romanian sections of the Prut River: Oroftiana (near the
entry, only including water level measurements), Radauti Prut, Stanca Aval (downstream),
Ungheni, Prisacani, Dranceni, Falciu, Oancea, and Sivita (which is directly influenced by the
Danube, so no data were collected from this station) (Fig. 3, Table 2). The first gauging
station was installed at Ungheni in 1914, and the newest station is Sivita, which was installed
in 1978. Much older water level and discharge data are available from stations in other places.
The data on the deviation of rainfall quantities were obtained from the Climate Prediction
Center NOOA and from the scientific literature (Hustiu, 2011).
**Table 2.** Morphometric data for the gauging stations on the Prut River (Romania)

| Gauging station | Inauguration year | Geographic coordinates | | River length from the confluence | Data on the catchment basin | | 0 m level of gauging station |
| --- | --- | --- | --- | --- | --- | --- | --- |
| | | Latitude | Longitude | km | Surface km$^2$ | Altitude m | mrBS (Meters Black Sea) |
| Oroftiana | 1976 | 48°11'12" | 26°21'04" | 714 | 8020 | 579 | 123.47 |
| Radauti Prut | 1976 | 48°14'55" | 26°48'14" | 652 | 9074 | 529 | 101.87 |
| Stanca Aval (Downstream) | 1978 | 47°47'00" | 27°16'00" | 554 | 12000 | 480 | 62.00 |
| Ungheni | 1914 | 47°11'04" | 27°48'28" | 387 | 15620 | 361 | 31.41 |
| Prisacani | 1976 | 47°05'19" | 27°53'38" | 357 | 21300 | 374 | 28.08 |
| Dranceni | 1915 | 46°48'45" | 28°08'04" | 284 | 22367 | 310 | 18.65 |
| Falciu | 1927 | 46°18'52" | 28°09'13" | 212 | 25095 | 290 | 10.04 |
| Oancea | 1928 | 45°53'37" | 28°03'04" | 88 | 26874 | 279 | 6.30 |
| Sivita | 1978 | 45°37'10" | 28°05'23" | 30 | 27268 | 275 | 1.66 |

Flood damage reports were collected from city halls in the Prut catchment and the
Inspectorate for emergencies in Botosani, Iasi, Vaslui, and Galati. In isolated areas, we
conducted our own field research. We note that some of the reports from city halls seem
exaggerated.

## 4 Results

The majority of floods in Romania are influenced by climate factors, manifesting at local and
European level (Birsan, 2015; Birsan and Dumitrescu, 2014; Birsan et al., 2012; Chendes et
al., 2015; Corduneanu et al., 2016). During the last decade of June (June 20, 2010) and the
end of July (July 30, 2010), a baroclinic area was localized in Northern Moldavia. This
favoured the formation of a convergent area of humidity. In this case, a layer of humid, warm
and unstable air was installed between the topographic surface and 2500 m of altitude. The
high quantity of humidity originitated from The Black Sea, situated 500 km away. The warm
tropical air is generated by the Russian Plain, overheated by a strong continentality climate.
The cold air from medium troposphere, inducted by the cut-off nucleum that generated
atmospheric instability, overlapped this structure of the low troposphere (Hustiu, 2011). The
synoptic context was disturbed by local physical-geographical factors, especially by the
orography of Eastern Carpathians, which led to extremely powerful heavy rains: e.g. 100-200
mm in 24 hours at the sources of Jijia (representing the amount that normally falls during June
and July) or 40-60 mm in 24 hours at the Romanian frontier with Ukraine and the Republic of
Moldova. The quantity of rainfall in 24 hours were 2-3 higher than the normal values for this
period (Hustiu, 2011) (Fig. 4).

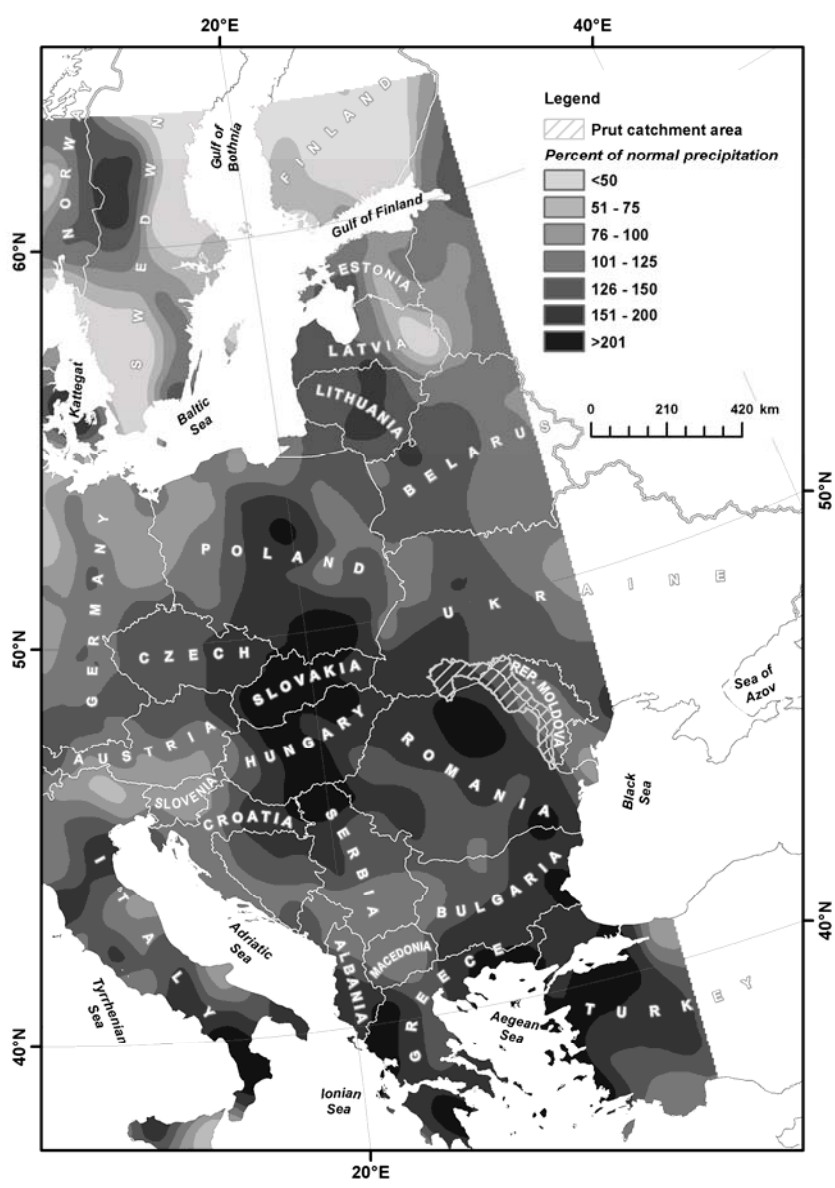

**Figure 4.** Deviation of monthly rainfall amounts (May-July 2010) from the yearly values -
Climate Prediction Center (source data: NOOA)
There were 6 main extremely rainy periods in Romania, especially in the Moldavian
hydrological basins (Prut and Siret): 21-23 June, 25-26 June, 28-30 June, 3-4 July, 6-7 July
and 9 July. Rainfall quantities recorded in June were higher. The flash floods registered in
Northern Moldavia in 28-29 June 2010 were generated by convective systems with slow
spreading. Even if the rainfalls from June 29th were lower, the floods had devastating effects
because they came on the context of the increasing water levels from 28 June 2010. The
climate convection was organized as a mesocyclone extended over Northern Moldavia (the
departments of Suceava and Botosani) (Hustiu, 2011).
Backwaters in the upper basins of the Prut and Siret (in northeast Romania) recorded
during the summer of 2010 were caused by atmospheric instability from 21 June-1 July 2010.
At this time, the flood danger level (CP) was exceeded on the Prut and Jijia Rivers. High
amounts of rain fell during three periods: 21-24 June 2010, 26-27 June 2010, and 28 June-1
July 2010. Precipitation exceeding 100 mm was recorded from 21-24 June (105 mm, at the
Oroftiana station) and from 28 June-1 July 2010 (206 mm at Padureni and 110 mm at Pomarla
on the Buhai River). Very high rainfall rates occurred within a brief timeframe: 51.5 mm/50
min. was recorded at Oroftiana station on the Prut River and 42.0 mm/30 min. at Padureni on
the Buhai River (Romanescu and Stoleriu, 2013a,b; Tirnovan et al., 2014b) (Fig. 5).
Precipitation in the Carpathian Mountains in Ukraine initiated a series of floods in the
upper Prut basin. Among the five flood peaks recorded by the Cernauti gauging station, we
noted one with a discharge of 2,070 $m^3$/s recorded on 9 July 2010 at 12:00. In comparison,
another flood recorded in May was not very high discharge value (308 $m^3$/s). In the
mountainous sector, the flood warning level (CA) was exceeded only twice, with water levels
of 523 cm (+25 cm CA) and 645 cm (+145 cm CA) (Fig. 6).
At the Oroftiana gauging station, where only the water levels are measured, the
flood danger level (CP) was exceeded four times, with levels of 716 cm (+66 cm CP), 743 cm
(+93 cm CP), 736 cm (+86 cm CP), and 797 cm (+147 cm CP, on 9 July 2010 at 12:00). The
flood warning level (CA) was exceeded throughout the entire flooding period (May-July
2010). In the month of May, the flood levels (CI) were not exceeded (Fig. 6). At the Oroftiana
gauging station, one registered solely the water levels data. And for all the other gauging
stations the discharge data are being registered, in addition to water level.

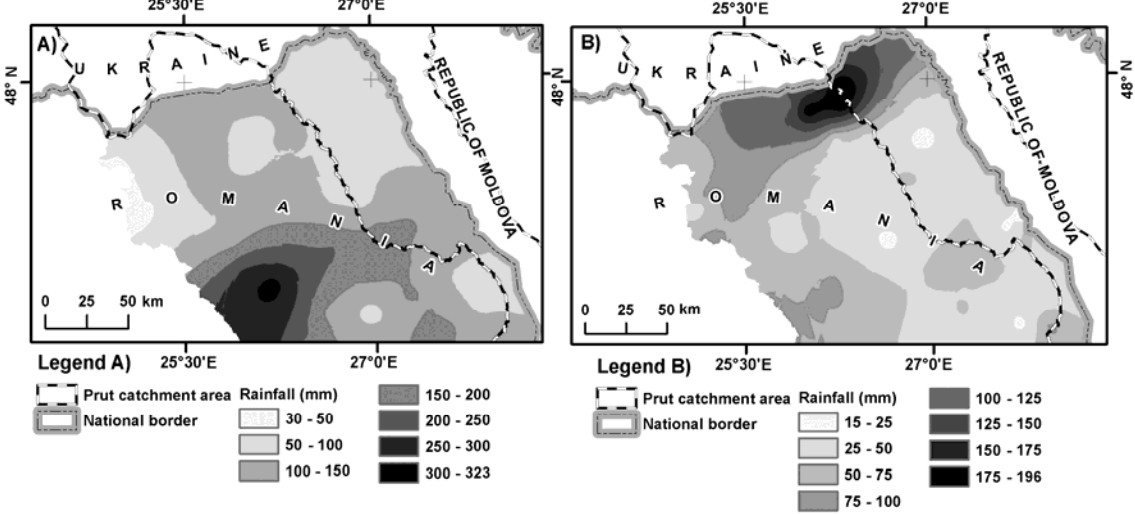

**Figure 5.** Cumulative precipitation amounts, in northeastern part of Romania, from 21-27
June 2010 (left) and 28 June-1 July 2010 (right)

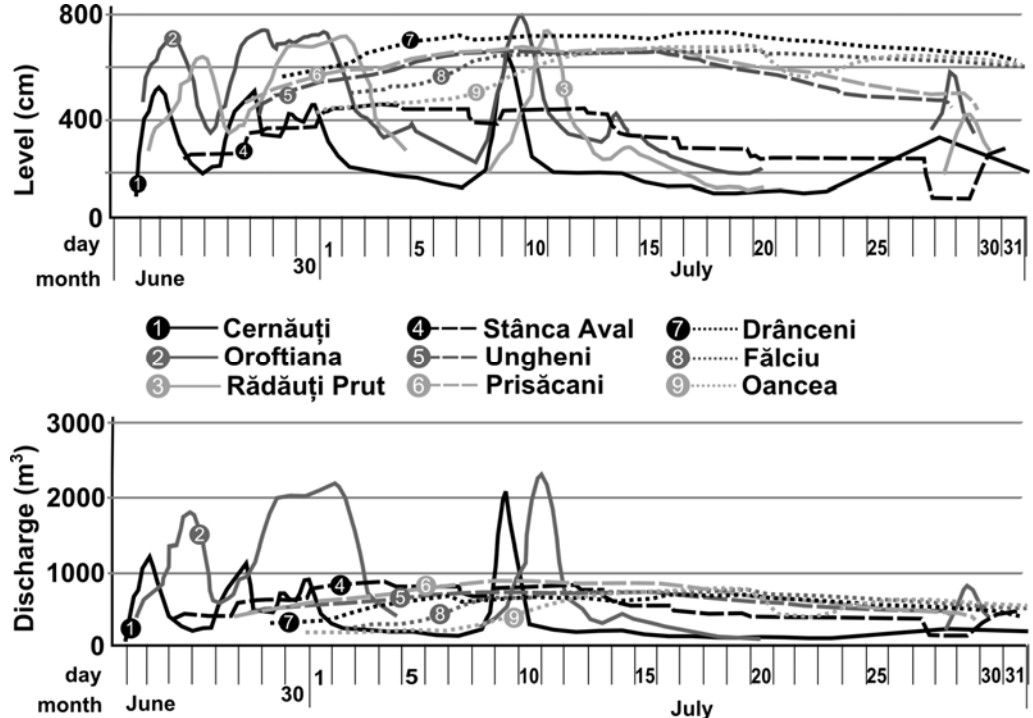

**Figure 6.** Water levels and discharge on the Prut River at the gauging stations of Cernauti, Oroftiana, Radauti Prut, Stanca Aval (downstream), Ungheni, Prisacani, Dranceni, Falciu, and Oancea during the summer of 2010

At the Radauti Prut gauging station, three important peaks were recorded on 26 June, 29 June-2 July 2010, and 10-11 July 2010. A maximum discharge of 2,310 $m^3$/s was registered on 10 July 2010 at 9 pm. The flood danger level (CP) was exceeded at four times, with water levels of 643 cm (+43 cm CP, on 25 June 2010), 685 cm (+85 cm CP, on 29 June 2010), 721 cm (+121 cm CP, on 29 June-2 July 2010), and 744 cm (+144 cm CP, on 10-11 July 2010) (Fig. 6).

The Stanca Aval (downstream) gauging station is controlled by overflow from the Stanca-Costesti reservoir. This control mitigates the flood hydrographs. The maximum discharge value at this station was 885 $m^3$/s on 3 July 2010. The flood level (CI) was exceeded from the beginning to the end of the flooding period. The flood danger level (CP) was exceeded from 1-13 July 2010, reaching a maximum water level of 460 cm (+85 cm CP, on 3 July 2010) (Fig. 6).

At the Ungheni gauging station, floods were recorded throughout the entire month of July. The maximum discharge was 673 $m^3$/s on 8 July 2010. Flooding continued until 5 August 2010. The flood danger level (CP) was exceeded during the 12-day period from 6-17 July 2010. The maximum water level was 661 cm (+1 cm CP) (Fig. 6).

Floods were also recorded throughout July at the Prisacani gauging station. The maximum discharge was 886 $m^3$/s on 9 July 2010. Flooding continued until 5 August 2010. The flood danger level (CP) was exceeded during the 16-day period from 4-19 July 2010. The maximum water level was 673 cm (+73 cm CP) (Fig. 6).

At the Dranceni gauging station, floods were recorded over a long period from the end of June until the beginning of August. The maximum discharge was 718 $m^3$/s on 17 July 2010. The flood danger level (CP) was reached or exceeded during the 18-day period from 4-22 July 2010. The maximum water level was 729 cm (+29 cm CP) (Fig. 6).

At the Falciu gauging station, floods occurred throughout July and during the first half
of August. The maximum discharge was 722 m³/s on 19 July 2010. The flood danger level
(CP) was reached or exceeded during the 35-day period from 6 July-2 August 2010. The
maximum water level was 655 cm (+55 cm CP) (Fig. 6).
At the Oancea gauging station, two backwaters were recorded in July and August.
The first backwaters on 19 July 2010 had a peak discharge of 697 m³/s and the second on 27
July 2010 had a peak discharge of 581 m³/s. Both backwaters exceeded the flood danger level
(CP) throughout the month of July. The maximum water level of the first backwater was 683
cm (+83 cm CP), and the maximum for the second was 646 cm (+46 cm CP) (Fig. 6).
Backwaters were caused by increasing water level of Danube River, which influences the
measurements results at the gauging stations situated on the downstream sector of Prut River.
The western tributaries of the Prut (within the Moldavian Plain) are numerous, but
they have only modest mean annual discharges. They are periodically affected by floods
following heavy summer rains. At the Stefanesti gauging station, within the downstream
sector of the Baseu River, floods were recorded from 1-4 July 2010. The maximum discharge
was 107 m³/s on 6 July 2010. The flood level (CI) was reached or exceeded for two days. The
maximum level was 355 cm (+5 cm CI) (Fig. 7). The Stefanesti gauging station is located in
the downstream sector of the dam and it is directly influenced by the discharge water from the
Stanca-Costesti Lake (since 1978).
At the Padureni gauging station on the Buhai River, two backwaters were recorded in
June and a secondary backwater in May. The maximum discharge was 470 m³/s on 28 June
2010. The flood danger level was exceeded during both backwaters, with water levels of 470
cm (+120 cm CP, on 28 June 2010) and 440 cm (+90 cm CP, on 29 June 2010) (Figs. 3, 7).
At the Todireni gauging station on the Sitna River (a tributary of the Jijia), floods
occurred from 1-4 July 2010. The maximum discharge was 19 m³/s on 1, 2, and 4 July 2010.
The flood level (CI) was exceeded on 1 and 2 July 2010. The maximum water level was 387
cm on 1 July 2010. The flood warning level (CA) was exceeded on 4 July 2010 (Figs. 3, 7).
At the Nicolae Balcescu gauging station on the Miletin River (a tributary of the Jijia),
floods were recorded from 26-29 June 2010. The maximum discharge was 60 m³/s on 6 June
2010. The flood level (CI) was exceeded just once, on 28 June 2010. The maximum level was
444 cm (+22 cm CI). The warning level (CA) was exceeded throughout the flooding period
(Figs. 3, 7).

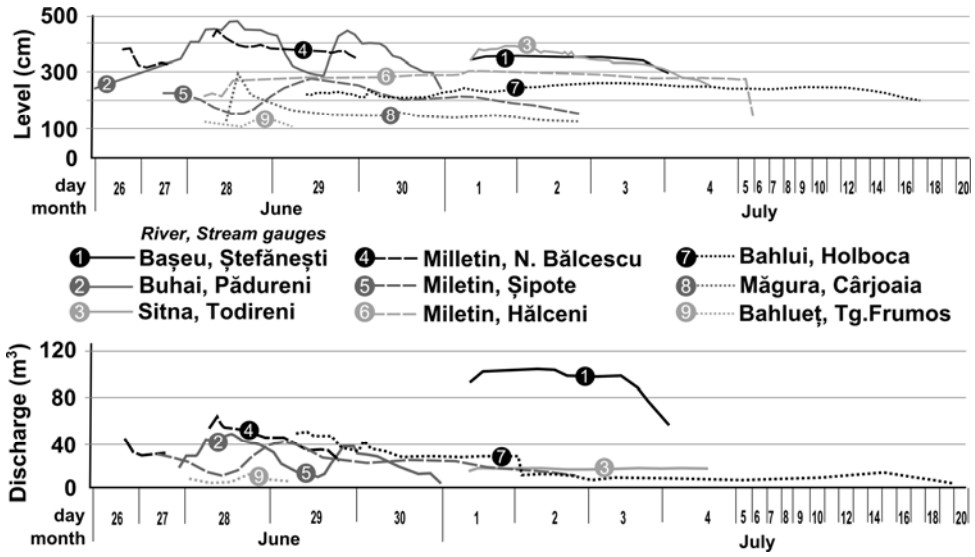

**Figure 7.** Water levels and discharge on the main Prut tributaries during the summer of 2010:
the Baseu, Buhai, Sitna, Miletin, Bahlui, Magura, and Bahluiet Rivers

At the Sipote gauging station on the Miletin, four backwaters were recorded from 22
June-2 July 2010. The maximum discharge was 45 $m^3$/s on 29 June 2010. The flood level (CI)
was exceeded from 29-30 June 2010. The maximum water level was 269 cm (+19 cm CI).
The warning level (CA) was exceeded throughout the flooding period (Figs. 3, 7).
At the Halceni gauging station on the Miletin, floods were recorded from 28 June-5
July 2010. The maximum discharge was 32 $m^3$/s on 1-2 July 2010. The flood danger level
(CP) was exceeded during the peak discharge period, with a water level of 302 cm (+2 cm
CP). The flood level (CI) was exceeded throughout the flooding period (Figs. 3, 7).
The Carjoaia gauging station on the Magura River (a tributary of the Bahlui), one
major backwater was recorded. The maximum discharge was 73.5 $m^3$/s on 28 June 2010. The
flood level (CI) was exceeded on 28 June 2010. The maximum water level was 280 cm (+90
cm CI) (Figs. 3, 7).
At the Targu Frumos gauging station on the Bahluet (atributary of the Bahlui), one
major backwater was recorded on 22 May 2010, with a maximum discharge of 48 $m^3$/s. The
flood danger level (CP) was reached on the same day and the maximum water level was 250
cm (0 cm CP). The flood warning level (CA) was exceeded throughout the flooding period
(Figs. 3, 7).
At the Harlau gauging station on the Bahlui (a tributary of the Jijia), successive and
increasing backwater were recorded from 22 May-1 July 2010. The maximum discharge was
32 $m^3$/s on 29 June 2010. The flood level (CI) was exceeded throughout the flooding period.
The maximum water level was 552 cm (+132 cm CI) (Figs. 3, 7).
At the Iasi gauging station on the Bahlui, floods occurred from 24 June-4 July 2010.
The maximum discharge was 44 $m^3$/s on 1 July 2010. The flood warning level (CA) was
exceeded throughout the flood. The maximum water level was 286 cm (+86 cm CA) (Figs. 3,
7).
At the Holboca gauging station on the Bahlui, floods were recorded from 29 June-17
July 2010. The maximum discharge was 50 $m^3$/s on 29 June 2010. The warning level (CA)
was reached or exceeded throughout the flooding period. The maximum water level was 259
cm (+59 cm CA) (Figs. 3, 7).
At the Dorohoi gauging station on the Jijia, several backwaters were recorded from 21
May-7 July 2010. The maximum discharge was 119 $m^3$/s on 29 June 2010. The flood danger
level (CP) was exceeded from 29-30 June 2010. The maximum water level was 760 cm (+160
cm CP). The flood warning level (CA) was exceeded throughout the flooding period (Figs. 3,
8).

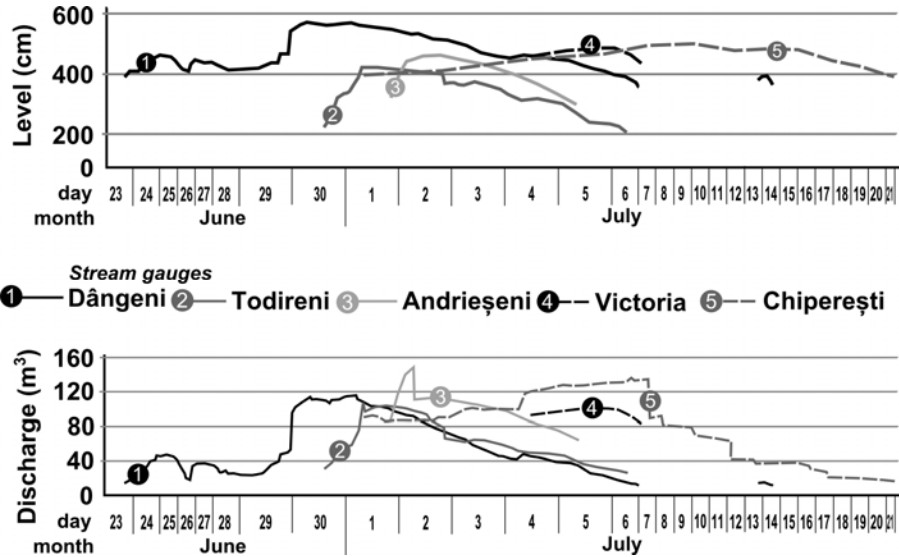

**Figure 8.** Water levels and discharge on the Jijia River at the gauging stations of Dangeni,
Todireni, Andrieseni, Victoria, and Chiperesti during the summer of 2010

At the Dangeni gauging station on the Jijia, several backwaters were recorded from 22
May-28 July 2010. The maximum discharge was 116 m$^3$/s on 1 July 2010. The flood level
(CI) was exceeded from 30 June-3 July 2010. The maximum water level was 578 cm (+108
cm CI). The flood warning level (CA) was exceeded throughout the flooding period (Figs. 3,
8).
At the Todireni gauging station on the Jijia, flooding occurred from 30 June-6 July
2010. The maximum discharge was 104 cm on 1 July 2010. The flood levels (CI) were
exceeded from 1-4 July 2010. The maximum water level was 417 cm (+47 cm CI). The flood
warning level (CA) was exceeded throughout the flooding period (Figs. 3, 8).
At the Andrieseni gauging station on the Jijia, flooding was recorded from 1-4 July
2010. The maximum discharge was 148 m$^3$/s on 2 July 2010. The flood danger level (CP) was
exceeded on 2 and 3 July 2010. The maximum water level was 461 cm (+11 cm CP). The
flood warning level (CA) was exceeded throughout the flooding period (Figs. 3, 8).
At the Chiperesti gauging station on the Jijia, successive and increasing backwaters
were recorded from1-19 July 2010. The maximum discharge was 136 m$^3$/s on 6 July 2010.
The flood warning level (CA) was exceeded throughout the flooding period. The maximum
water level was 497 cm (+97 cm CA) (Figs. 3, 8).
At the Victoria gauging station on the Jijia, flooding occurred from 4-7 July 2010. The
peak discharge was 100 m$^3$/s on 5 July 2010. The flood warning level (CA) was exceeded
throughout the flooding period. The maximum water level was 485 cm (+35 cm CA) (Figs. 3,
8).
At the Capitanie A.F.D.J. gauging station on the Danube, record floods occurred. The
maximum discharge was 16,300 m$^3$/s on 5-6 July 2010, which is a historic discharge for the
Galati station. The flood level (CI) was exceeded from 26 June-14 July 2010 (Fig. 9).

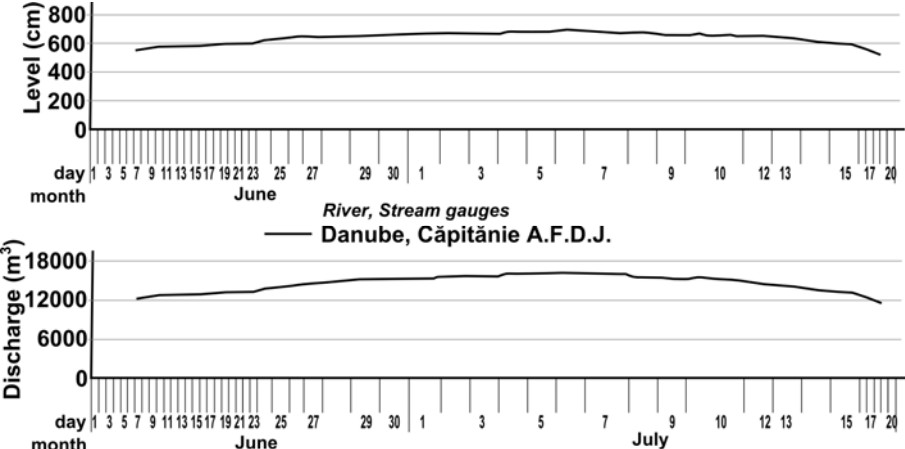

**Figure 9.** Water levels and discharge on the Danube at the Capitanie A.F.D.J. gauging station in the summer of 2010

## 5 Discussion

Cumulative heavy rains from 21-24 June, 26-27 June, and 28 June-1 July 2010 caused water levels to exceed the flood danger level (CP) by 40-150 cm on the Prut in the Oroftiana-Radauti Prut sector and by 30-150 cm in the upper basin of the Jijia. The flood level (CI) was exceeded by 80-110 cm in the middle basin of the Jijia and in its tributaries (Sitna, Miletin, and Buhai). Discharges within the lower Jijia basin were controlled by upstream reservoirs and downstream polders in the lower reaches of the Jijia.

The Oroftiana gauging station only records water level measurements. The Radauti Prut gauging station may be influenced by the water stored in the Stanca-Costesti reservoir (which occurred during the historic flood of 2008) (Romanescu et al., 2011a,b). The Stanca downstream gauging station may be influenced by overflow from the Stanca-Costesti reservoir. The Oancea gauging station, situated near the mouth of the Prut, may be influenced by waters from the Danube. The water level registered at the Radauti Prut gauging station could have been influenced by the backwaters caused by Stanca-Costesti Lake. The most obvious case of backwaters was registered during the 2008 historic flood.

High discharge and water levels of 2,310 m³/s and 744 cm (+144 cm CP), respectively, were recorded at the Radauti Prut gauging station. The 2010 values are remarkable lower than the maximum values recorded in 2008 of 7,140 m³/s and 1,130 cm (+530 cm CP) (the highest value for Romanian rivers). This value was recalculated after two years (through recomposed discharges), resulting in a discharge of 4,240 m³/s, which is the second highest value in Romania (after the historic discharge of 4,650 m³/s on the Siret in 2005) (Romanescu et al., 2011a,b). The existence of five backwater peaks (with the second and third backwaters being weaker) clearly indicates that they were caused by heavy rains in the Carpathian Mountains in Ukraine. A volume of 200-400 mm of rainfall (ie 50-80% of the annual amount) was recorded between 1 May and 15 July 2010. During the flood manifested in 2008, a historic discharge value was registered for Prut River, but the by-passed water volume was low (in upstream of Stanca-Costesti dam) because the flood duration was short. The 2010 flood registered lower maximum discharges compare to 2008, but it by-passed a larger water volume, as flood lasted longer.

The flood hydrographs recorded at the Stanca Aval (downstream) gauging station features flattened and relatively uniform backwaters, mostly in the central part of the river.

This behaviour is due to the influence of Stanca-Costesti reservoir, which significantly
reduced the maximum discharge at Stanca Aval (885 m³/s) compared to the Radauti Prut
gauging station upstream of the reservoir. The water level was maintained within the upper
limit recorded by longitudinal protection dams.

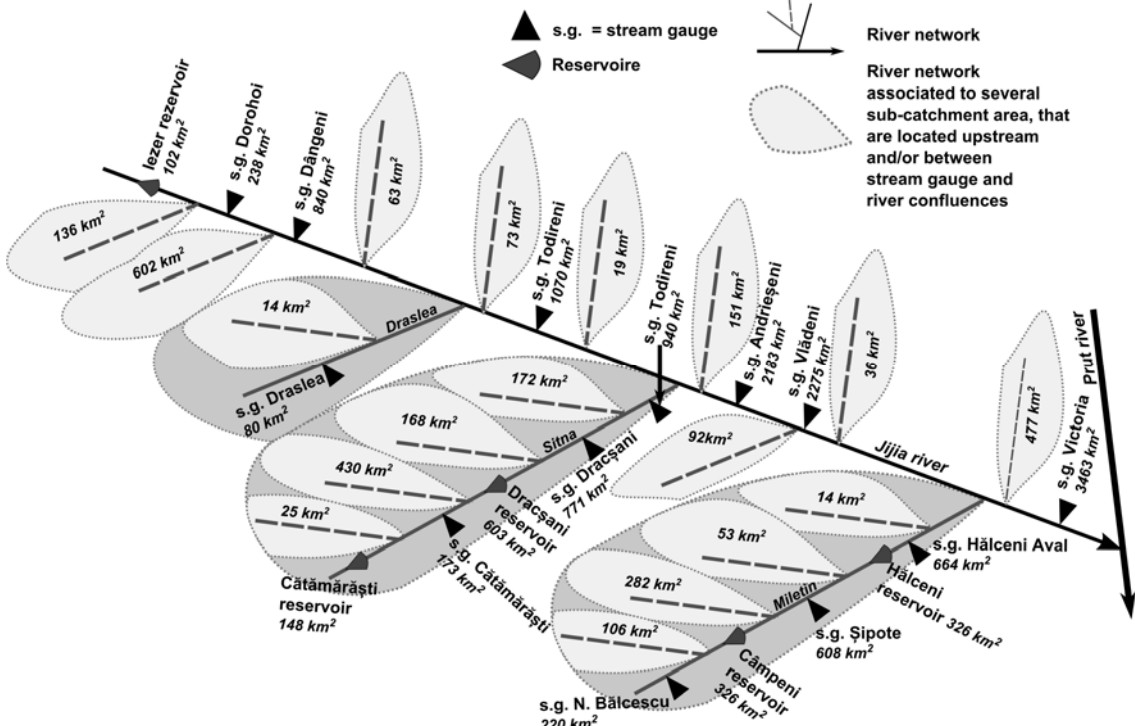

**Figure 10.** Distribution of sub-basins within the Jijia catchment and placement of the main
ponds

The Ungheni, Prisacani, Dranceni, and Falciu gauging stations had a flattened and
uniform backwater, which signifies upstream control, including some of the tributaries. The
flood danger level (CP) was exceeded by a few centimetres and the floodplain was partially
flooded in these areas. The high discharges recorded at the Prisacani station occurred because
of waters in the upper Prut basin, including controlled spills from the Stanca-Costesti
reservoir. Downstream of the Prisacani station, the influence of the Jijia becomes obvious: it
increases the water level and lengthens the duration of floods.
Stronger floods within the middle reaches of the Prut occur because of its tributaries.
Flooding on the Baseu, Sitna, Miletin, Jijia, Bahluet, and Bahlui Rivers was strong, but it was
mitigated for the most part by the existence of ponds (Fig. 10). Therefore, the excess water
entering Romania from Ukraine entered the Stanca-Costesti reservoir. The excess water
downstream of the Stanca-Costesti reservoir came from tributaries. Discharge from the
tributaries is controlled by hydrotechnical works within each tributary's catchment. The Jijia
and Bahlui catchments are 80% developed. The water levels downstream of these tributaries,
in the lower reaches of the Prut, are mitigated by the extreme width of the Prut floodplain (the
most important wetland of the interior Romanian rivers).
The system of polders in the lower reaches of the Jijia served as an effective trap for
surplus water. High discharges on the Danube, which reached a historic maximum of 16,300
m³/s at Galati (July 5th, 2010), would have flooded the city centre without the precincts
constructed on the Jijia that stopped a portion of the floodwaters. When the floods on the
Danube ceased, the water was gradually eliminated from the polders, which explains why
high water levels persisted in the lower Prut for a long time (Fig. 11).

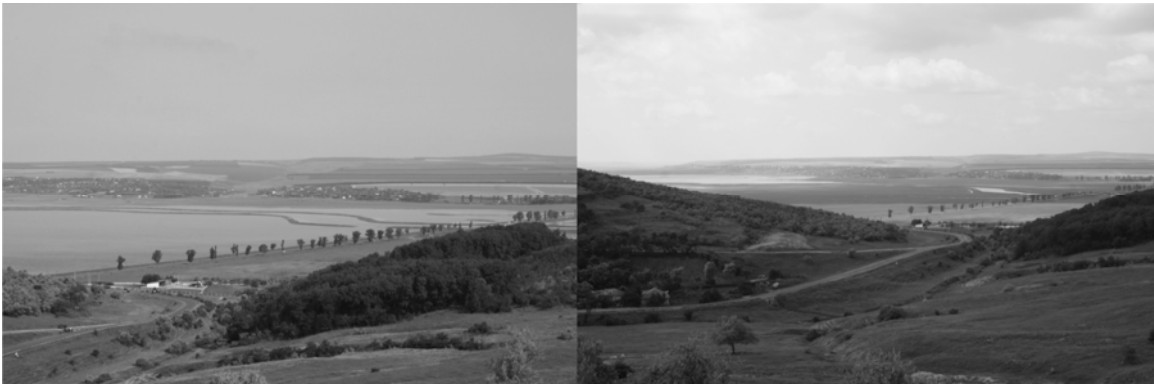

**Figure 11.** Polders on the Jijia and the floods recorded in the summer of 2010: storage of
excess water (left) and its elimination (right)

Discharge at the Oancea gauging station increased dramatically from 4-5 July 2010,
coinciding with the increased discharge on the Danube at Galati. The backwater at Oancea
was also enhanced by backwater from the Danube. The second backwater was caused by
upstream contributions. The flood danger level (CP) at Oancea was exceeded by +83 cm (CP)
during the first backwater and by +46 cm (CP) during the second backwater (Table 3). The
discharge increase and the historic values registered were caused by several factors, such as:
the water input from the upstream sector of Prut River and the water input added by the
Danube backwaters.

**Table 3.** Values of CA, CI, and CP for the Oancea (Prut) and Galati (Danube) gauging
stations.

| Gauging station | CA (Warning level) | CI (Flood level) | CP (Danger level) |
|---|---|---|---|
| Oancea (Prut) | 440 | 550 | 600 |
| Galati (Danube) | 560 | 600 | 660 |

The city of Galati is situated at the confluence of the Prut and the Danube Rivers.
Thus, water at the Oancea station may be influenced by the Danube and the Prut. In the
summer of 2010, the highest values of discharge and water level at Galati were recorded
(Tables 4, 5). The control of flooding on the Prut meant that floodwaters in Galati reached the
sector of banks where flood infrastructure had been developed (the sea-cliff) as well as the
lower areas of the city (Fig. 12).

**Table 4.** Maximum water levels during flooding in the summer of 2010 for the Danube
compared to values from other flood years.

| River | Gauging station | Maximum levels in the year (cm) | | | | |
|---|---|---|---|---|---|---|
| | | 2010 | 2006 | 2005 | 1981 | 1970 |
| Danube | Galati | 678 | 661 | 600 | 580 | 595 |
| | Isaccea | 537 | 524 | 481 | 490 | 507 |
| | Tulcea | 439 | 437 | 399 | 415 | 429 |

**Table 5.** Maximum discharges during flooding in the summer of 2010 for the Danube
compared to the maximum values from 2006.

| River | Gauging station | Maximum discharges in the year (m$^3$/s) | |
|---|---|---|---|
| | | 2010 | 2006 |
| Danube | Galati | 16300 | 14220 |
| | Isaccea | 16240 | 14325 |
| | Tulcea | 6117 | 5768 |

Discharges and water levels in the middle sector of the Prut River (recorded at the Oroftiana, Radauti Prut, and Stanca Aval stations) rank third in the hierarchy of floods (after 2008 and 2005). Values for the tributaries (particularly the Jijia, Buhai, Miletin, and Sitna) rank first in the hierarchy of floods (Table 6).

**Table 6.** Maximum water levels during flooding in the summer of 2010 compared to 2008 and 2005.

| River | Gauging station | Maximum level cm | Day | Hour | Difference from the three levels of danger Cm | Maximum level 2008 cm | Maximum level 2005 cm |
|---|---|---|---|---|---|---|---|
| Prut | Oroftiana | 717 | 24.06 | 11 | +67 CP | 867 | 703 |
| | | 744 | 28.06 | 11-12 | +94 CP | - | - |
| | | 737 | 1.07 | 04 | +87 CP | - | - |
| | | 797 | 9.07 | 17-18 | +147 CP | - | - |
| | | 425 | 13.07 | 20 | +75 CA | - | - |
| Prut | Radauti Prut | 643 | 25.06 | 18-19 | +43 CP | 1130 | 680 |
| | | 686 | 29.06 | 17 | +86 CP | - | - |
| | | 722 | 1.07 | 23 | +122 CP | - | - |
| | | 744 | 10.07 | 19-20 | +144 CP | - | - |
| Prut | Stanca Downstream | 461 | 3.07 | 15-22 | +86 CP | 512 | 331 |
| Jijia | Dorohoi | 750 | 29.06 | 09 | +150 CP | 558 | 646 |
| | | 722 | 30.06 | 05 | +122 CP | - | - |
| | | 630 | 30.06 | 17 | +30 CP | - | - |
| Jijia | Dangeni | 575 | 30.06 | 08 | +105 CI | 449 | 512 |
| | | 579 | 1.07 | 05 | +109 CI | - | - |
| Jijia | Todireni | 417 | 1.07 | 08 | +77 CI | 123 | 420 |
| Buhai | Padureni | 470 | 28.06 | 19-20 | +120 CP | 292 | - |
| Miletin | Nicolae Balcescu | 444 | 28.06 | 15 | +24 CI | 286 | 334 |
| Miletin | Sipote | 226 | 27.06 | 12 | +76 CA | 198 | 236 |
| | | 269 | 29.06 | 18 | +19 CI | - | - |
| Miletin | Halceni | 302 | 1.07 | 15-18 | +2 CP | 226 | 238 |
| Sitna | Todireni | 378 | 1.07 | 17 | +28 CI | - | - |

The floods recorded in the summer of 2010 in the Buhai catchment (a tributary of the Jijia, which is a tributary of the Prut) caused backwaters to emerge at the mouth of the river. The manifestation of this backwater phenomenon is unique because the floodwaters of the Buhai River climbed the Ezer dam (on the Jijia River) and flooded its lacustrine cuvette. The phenomenon was named "spider flow" (Romanescu and Stoleriu, 2013a,b) (Fig. 13).

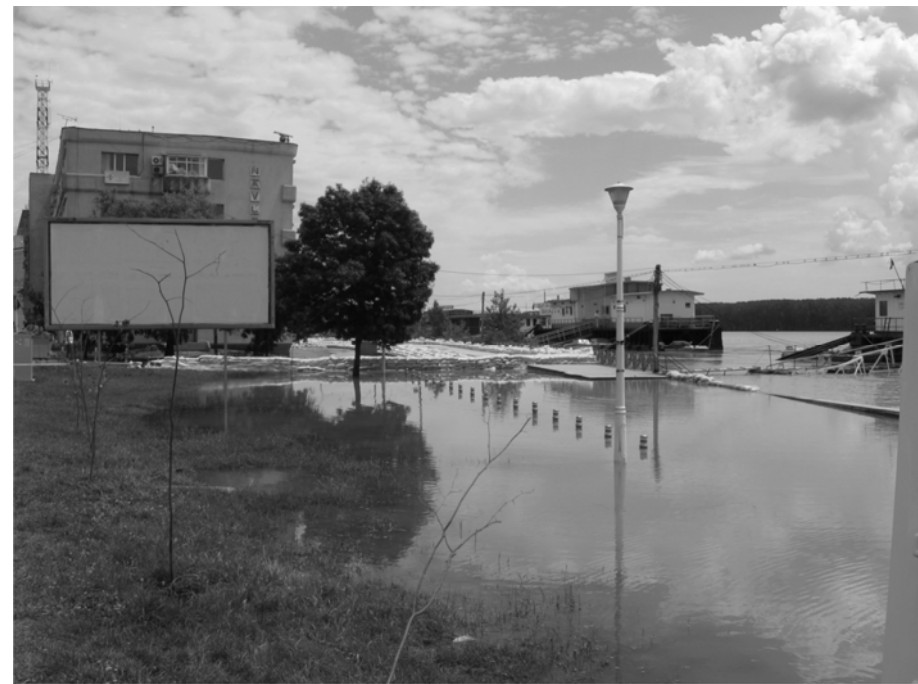

**Figure 12.** Flooding of the sea-cliff and the NAVROM headquarters in Galati

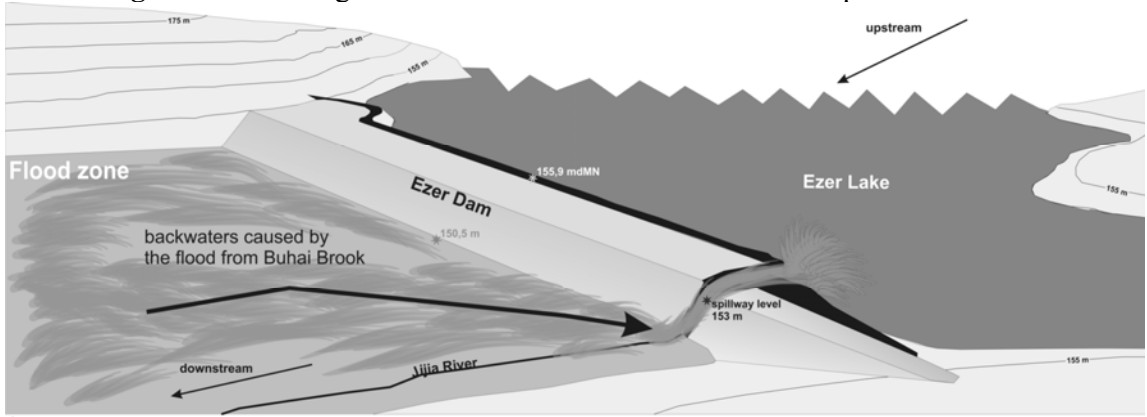

**Figure 13.** The "spider flow" phenomenon in which the Buhai waters climbed the Ezer dam
on the Jijia, in the area of confluence of the two rivers

## 6 Conclusions

In the summer of 2010, large amount of precipitation occurred in Central and Eastern Europe. Heavy rains in northeast Romania caused devastating floods in the Prut and Siret basins. Romania incurred huge economic damages. The flooding in 2010 was comparable with previous strong flood years in 2005, 2006, and 2008 in Romania. The greatest damage occurred in, and the most arable area was destroyed in, the middle Prut basin in the Jijia-Bahlui Depression of the Moldavian Plain.

Discharge in the downstream sector of the Prut was controlled by the Stanca-Costesti reservoir, which ranks 2$^{nd}$ in Romania in terms of active reservoir volume (1,400 million m$^3$, after the Iron Gates I, with 2,100 million m$^3$). It has a surface area of 5,900 ha for a NRL. Under normal circumstances, the Stanca-Costesti reservoir can retain enough water to control the downstream discharge and water level. The provision of an attenuation water

volume (550 million m$^3$) within the lake basin is efficient in retaining a 1% probability flood (reducing it from 2,940 m$^3$/s to 700 m$^3$/s). Together with the embankments located on the dam downstream sector, it helps preventing the flooding of 100,000 hectares of meadow. At a normal retention level, Stanca-Costesti Lake has a total area of 5,900 ha and a water volume of 1.4 billion m$^3$.

Discharges downstream of the Stanca-Costesti reservoir are controlled by reservoirs and retention systems constructed on the main tributaries of the Prut. We emphasize that the Jijia and Bahlui catchments have hydrotechnical works on 80% of their surface areas. The system of polders in the downstream sector of the Jijia River was used extensively to mitigate discharge and prevent the city of Galati from flooding (Galati is the largest Danubian port, situated at the confluence of the Prut and the Danube Rivers).

The gauging stations in the lower sector of the Prut recorded high discharges and water levels because of excess water coming from upstream (the middle sector of the Prut). At the Oancea gauging station, however, which is situated near the discharge of the Prut into the Danube, there is a significant backwater influence. The Danube had historic discharge at Galati, which affected the water level at Oancea station on the Prut.

Floods during the summer of 2010, in northeast Romania, rank third among hydrological disasters in Romanian history after the floods of 2005 and 2008, which also occurred in the Siret and Prut catchments. The 2010 floods caused grave economic damage (almost one billion Euros in just the Prut catchment) and greatly affected agriculture. Furthermore, six people died in Dorohoi, on the Buhai River.

The 2010 floods caused a unique backwater phenomenon at the mouth of the Buhai River. Floodwaters from the Buhai climbed the Ezer dam (situated on the Jijia River) and flooded its lacustrine cuvette. The phenomenon was called "spider flow". In order to avoid such phenomena it is necessary to increase the height of the overflow structure.

*Acknowledgments.* This work was supported by the Partnership in Priority Domains project PN-II-PT-PCCA-2013-4-2234 no. 314/2014 of the Romanian National Research Council, called "Non-destructive approaches to complex archaeological sites. An integrated applied research model for cultural heritage management" – arheoinvest.uaic.ro/research/prospect. The authors would like to express their gratitude to the employees of the Romanian Waters Agency Bucharest, Siret Water Administration Bacau, particularly to Jora Ionut, PhD, a hydrologist within this research and administration agency, who was kind enough to provide a significant part of the data used in the present study.

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
