# Peer review of "Exceptional floods in the Prut basin, Romania, in the context of heavy rains in the summer of 2010"

_Natural Hazards and Earth System Sciences, 2016_

## Referee Comment (RC1) · Anonymous Referee #1 · 21 Nov 2016

General comment: The paper addresses a very interesting report about floods that occurred in 2010 in Romania. It is well documented with respect to water levels and discharges observed in the trans boundary Prut basin (nearly 27000 km$^2$). The paper might be considered as a reference document about largest floods in Europe. The described river system is complex because of existing dams, ponds and the particular geometry of river confluences. However the geometric data about the river system are not well described here.

Unfortunately, the paper organization is not easy to understand. The reader who is not used with this river system cannot follow the presentation. Detailed comments are as following. Line 34 "Floods are one of the most important natural hazards on Earth"

references are about Europe and not the earth Line 36. "Significant funds...". You may cite the date provided in Merz et al. http://www.nat-hazards-earth-syst-sci.net/nhess-special_issue77-preface.pdf The reference in lines 37 to 44 should be documented and separated in different topics. Effectively the list is too long and is a mixing of several subjects. For example: -Ahilan et al. 2012 is about statistical distribution of maximum annual discharge using GEV and relationships with basin geology - Alfieri et al. 2015 is about climate change impacts on floods - Berariu et al. 2015 is about the effects of disasters on infrastructures such as transportation infrastructures and their interdependence, etc... Line 61: are the Stanca-Costesti reservoir and the Prut reported in Fig. 1? Line 83 altitude in the catchment Line 90 Jijia basin area is not documented while this basin is important in the last part of the paper. Line 94 what is the criteria to define a "large pond"?. Line 111 "measurements were taken to estimate the discharge." It is important to say which kind of measurements. Lines 113 to 118 Same remark as in lines 37 to 44. It should be clear what type of method is behind a given reference. For example Ali et al. (2012) used tracers while Delli-Priscoli and Stakhiv examined "the performance of existing flood protection systems". Line 132 did CA, CI, CP have been defined before? Line 148, 149 the methodology should be more detailed. Line 154 and 164 are not compatible (1 July, 9 July). Line 168 You need to specify what is registered in each station. What do you mean by "only water levels"? the stations reported in Table 1 should be easily identified in Fig. 3 (by using a different marker) and what is observed (level or discharge should be mentioned. Fig. 5 is not easy to read Line 199 and line 203. What is meant by "floods were recorded"? Do you mean that a flood gauging was operated instead of using the rating curve? Line 211 the peculiarity of Oancea gauging station and Sivita station distinguishing tidal effects should be documented. Line 243 and elsewhere "Fig. 3 and 6" is not clear. Fig 6 is not easy to read. The peculiarity of Stefanesti(?) station should be mentioned and analyzed in the text. (lines 218 to 221) In all figures, with levels and discharges plts the basin area should be mentioned as well as in lines 310-315. Line 316. In is not clear why this mention here "The Oroftiana gauging station only records water level

measurements." Idem until line 321. What is the consequence on data accuracy? Line 317. Why this influence? Lines 329 – 330. Was rainfall observed? Line 331-341 should in the study area section Line 371; When did this record happened? Line 380. Is this increase a result from what was said before? Line 386 Table 2 should be in the study area section Line 412 the backwater phenomena are effectively very difficult to assess and to predict. Lines 427 to 432. The role of the reservoir and its location in comparison to the river stations is not well described in the text. Line 449 Fig.12 presents challenging issues for water management. Information lines 453 to 455 are of great importance and should be reported very early in the text corp.

---

## Referee Comment (RC2) · Anonymous Referee #2 · 27 Nov 2016

General comments

The paper copes with the exceptional floods that hit Central Europe and particularly Romania in summer 2010. The work shows interesting flood data for the examined area (though partially presented by the authors in previous works), but it does not constitute a clear contribution to the understanding of these phenomena in the Prut basin, also for its complicated river network. In fact, though the work contains a lot of information on water levels and discharges observed during huge floods, these are mainly ranked values, roughly compared to similar past events but not statistically defined. In other terms, the paper is too much focussed to the simple inventory of flood values in several gauge stations, and poor attempts to link them to physical reasons or to proba-

bilistic interpretation have been made by authors. Thus, the readability of the paper is not good enough, mainly in the paragraph of the results.

Specific comments

Specifically, though the paper is mainly devoted to flood events, the context of heavy rains of the summer of 2010 (as in the title) is poorly described and could be largely improved. This could be done, for example, by coupling flood diagrams with rainfall histograms, when possible, or by comparing cumulative rainfall values recorded in this event with rainfall that caused other historical floods (also cited in the work). Anyway, the main drawback of the paper is the weak connection between rainfall and floods. In fact, though the period claimed as characterized by intense rainfall is 21 June -1 July 2010, a long set of summer flood (or water level) values is offered to the reader, neither providing any kind of link with triggering precipitation, nor any estimation of the return periods of the rainfall or flood values. Actually, the results are only described by means of simple ranks among critical events. To improve the paper, the paragraph devoted to the results should present at least some evaluations on the estimated frequencies (and not only on critical cases) of the flood values, thus providing more statistical sound to the work. On the other side, the interesting information on water stages and floods overcoming the specific thresholds is described too simply. The valuable data base can be better employed, for example, by combining the temporal overcoming of the higher thresholds in the flood diagrams with the occurrence of the main damages and casualties. This could also provide material for a further interesting discussion on false and missing alarms in the Prut River. Moreover, the work suffers from too much citations, not everywhere appropriate, and from figures affected by some inaccuracies. In brief, though well documented as regards the discharge values, the structure of the work is disorganised enough, with a scarce employment of statistical methodologies and a long section devoted to the results, which consist principally in a list of flood values, with no link to occurrence frequencies. As a result, the scientific approach of the work is not statistically accurate. Thus, a substantial revision of the paper is needed

to improve the quality of the work and provide effectiveness to the flood analysis of the 2010 event in the Prut River.

Technical corrections

Line 18: avoid the word "etc." in the abstract; line 34: change "Earth" with "earth". Lines 61, 153 (and others): I don't understand if the authors use properly the terms "tidal bore" in rivers, except in the case of backwaters actually induced by reservoirs or confluences. Try to be more accurate. Line 76, Figure 2, legend: change "Km" with "km"; avoid decimal ciphers in elevation values. Line 83: it's not clear why the mean altitude assume different values. Line 84: from the figure, the maximum width of Prut basin seems not to be 30 km (even in the lower reaches). Improve the sentence. Lines 101-103: the sentence is trivial (except, maybe, for the presence of the several ponds, which should be recalled). Anyway, the differences among the discharges for the various sections seem very small for such a large river. Lines 107-118: The cited methodologies are not useful for analysing floods, but for recording and collecting data. The paragraph contains too much references and not all perfectly focussed on the issue. The sentence needs a better explanation. Lines 126: it's not usual the call to the Berg intensity scale. If possible, add a reference. Line 132: the CA, CI and CP flood threshold levels should be clearly defined. Line 141: change "1915" with "1914", as noted in the table 1. Line 144, Table 1: the parameter "0 mira level", and mainly its unit "mrBS", should be better explained (or changed). Line 165: the use of the term "significant" should be associated to statistical analysis. Line 175, Figure 4: the values in the legend should not show decimal ciphers. Line 175, Figure 4: can the areal extension of the rainfall analysis be enlarged to the whole Prut basin? Line 177, Figure 5: it's useless to span the graphs before and after the period 20 June – 31 July, that could be better centered with no temporal amplification. Line 235, figure 6: it's not useful to extend the graphs after 1 July. Line 274, figure 7: the temporal amplification can be easily avoided. Line 303, figure 8: the temporal amplification is useless. The legend ("X scale, 0-24 hours") has no meaning. Line 324: the term "significantly"

should be associated to statistical analysis. Lines 325-327: the sentence "This value was recalculated ..." should be better explained. Line 339: the sentence "... allowing the mitigation of 1% ..." is not clear. Line 373: there are some words repeated ("was eliminated gradually"). Line 430: it can be used directly the acronym "NRL", previously defined in line 335. References in Romanian language should report the words "(in romanian)" at the end of the citation.

———————————————————

---

## Author Comment (AC1) · 1 Dec 2016

Dear referee, thank you for your interests about our article,

Referee comment 1: Line 34 "Floods are one of the most important natural hazards on Earth" references are about Europe and not the earth

Authors' answer 1: Concerning line 34 we omitted to detail the phrase from "Floods are one of the most important natural hazards on Earth" to " Floods are one of the most important natural hazards in Europe (Thieken et al., 2016) and on earth as well (Merz et al., 2010; Riegger et al., 2009). They generate major human life losses and property damage (Wijkman and Timberlake, 1984).", and we modified in text's paper.

[Figure]

Referee comment 2: Line 36. "Significant funds...". You may cite the date provided in Merz et al. http://www.nat-hazards-earth-syst-sci.net/nhess- special_issue77-preface.pdf

Authors' answer2 : We summarized the ideas specified by Merz et all. Into next paragraph: "According to Merz et al. (2010) "the European Flood Directive on the assessment and management of flood risks (European Commission, 2007) requires developing management plans for areas with significant flood risk (at a river basin scale), focusing on the reduction of the probability of flooding and of the potential consequences to human health, the environment and economic activity." (p. 511)."

Referee comment 3: The reference in lines 37 to 44 should be documented and separated in different topics. Effectively the list is too long and is a mixing of several subjects. For example: -Ahilan et al. 2012 is about statistical distribution of maximum annual discharge using GEV and relationships with basin geology - Alfieri et al. 2015 is about climate change impacts on floods - Berariu et al. 2015 is about the effects of disasters on infrastructures such as transportation infrastructures and their interdependence, etc...

Authors' answer3: We rephrase the paragraph about references between lines 37-44 Several studies investigated catastrophic floods or the floods that generated significant damage. They focused on: the statistical distribution of maximum annual discharge, using GEV and the links with the basin geology (Ahilan et al., 2012); climate change impacts on floods (Alfieri et al., 2015; Detrembleurs et al., 2015; Schneider et al., 2013; Whitfield, 2012); disasters effects on infrastructures such as transportation infrastructures, and their interdependence (Berariu et al., 2015); historical floods (Blöschl et al., 2013; Strupczewski et al., 2014; Vasileski and Radevski, 2014) and their links to heavy rain (Bostan et al., 2009; Diakakis, 2011; Prudhomme and Genevier, 2011; Retsö, 2015); public perceptions of flood risks (Brilly and Polic, 2005; Feldman et al., 2016; Rufat et al., 2015); land use changes and flooding (Cammerer et al., 2012); the evolution of natural risks (Hufschmidt et al., 2005); geomorphological effects of

floods in riverbeds (Lichter and Klein, 2011; Lóczy and Gyenizse, 2011; Lóczy et al., 2009, 2014; Reza Ghanbarpour et al., 2014); the spatial distribution of floods (Moel et al., 2009; Parker and Fordham, 1996); the interrelation between snow and flooding (Revuelto et al., 2013).

Referee comment 4: Line 61: are the Stanca-Costesti reservoir and the Prut reported in Fig. 1?

Authors' answer4: We modified the Figure 1, in order to appear River Prut, Danube and Stanca-Costesti reservoir.

Referee comment 5: Line 83 altitude in the catchment

Authors' answer5 : The situation observed at line 83 is an unfortunate manner of writing for describing the mean altitude within Prut catchment basin. The phrase was adjusted as follow: "The mean altitude of the midstream sector of catchment area is 130 m, and for the downstream sector is 2 m."

Referee comment 6: Line 90 Jijia basin area is not documented while this basin is important in the last part of the paper.

Authors' answer6 : We introduced some detailed information concerning Jijia River: "Jijia River has 275 km in length, a catchment area of 5757 km2 and an annual average flow of 14 m3/s. Its most important tributaries are Miletin, Sitna and Bahlui."

Referee comment 7: Line 94 what is the criteria to define a "large pond"?

Authors' answer7 : Small ponds are used as drinking water for livestock or to irrigate subsistence rural households. They usually belong to individual households. Large ponds on the other hand have multiple uses, such as: flooding mitigation, irrigation, fish farming etc. They resisted better in time because of their significant surfaces and depths. These large ponds belong to rural or urban communities.

Referee comment 8: Line 111 "measurements were taken to estimate the discharge."

It is important to say which kind of measurements.

Authors' answer8: Mathematical methods were used to reconstitute discharges and terrain measurements using land surveying equipment (Leica Total Station) were used to calculate the surface of the stream cross-section.

Referee comment 9: Lines 113 to 118 Same remark as in lines 37 to 44. It should be clear what type of method is behind a given reference. For example Ali et al. (2012) used tracers while Delli-Priscoli and Stakhiv examined "the performance of existing flood protection systems". Line 132 did CA, CI, CP have been defined before?

Authors' answer9 : We restructured the paragraph such as: "The recording and analysing methodology used is standard or slightly adapted to local conditions: e.g. the influence of physical-geographical parameters on runoff (Ali et al., 2012; Kappes et al., 2012; Kourgialas et al., 2012; Waylen and Laporte, 1999); the management of risk situations (Delli-Priscoli and Stakhiv, 2015; Demeritt et al., 2013; Grobicki et al, 2015 Grobicki et al, 2015); the role of reservoirs in flood mitigating (Fu et al., 2014; Serban et al., 2004; Sorocovschi, 2011); the probability of flooding and the changes in the runoff regime (Hall et al., 2004, 2014; Jones, 2011; Seidu et al., 2012a,b; Wu et al., 2011); flood prevention (Hapuarachchi et al., 2011); runoff and streamflow indices (Nguimalet and Ndjendole, 2008); morphologic changes of riverbeds or lake basins (Rusnák and Lehotsky, 2014; Touchart et al., 2012; Verdu et al., 2014) etc."

Referee comment 10: Line 148, 149 the methodology should be more detailed.

Authors' answer10 : The cartographic basis used to map altitudes and slopes is Shuttle Radar Topography Mission (Global Land Cover Facility, 2016), at a 1:50000 scale. The vector layers were projected within a geodatabase, using ArcGis 10.1. They include stream lines, sub-catchment basins, and reservoirs and ponds polygons, as well as gauging station points. In order to generate the GIS layers, we applied the following methods: digitisation, queries, conversion, geometries calculation (length, surface) and spatial modelling. Water levels and discharges data were processed and plotted on

charts using the Open Office software. We also used the Inkscape software to design the final maps and images.

Referee comment 11: Line 154 and 164 are not compatible (1 July, 9 July).

Authors' answer11 : In the first case it's about rainfalls registered in Romania (on July 1st) and in the second case it's about those registered in Ukraine (on July 9th).

Referee comment 12: Line 168 You need to specify what is registered in each station. What do you mean by "only water levels"? the stations reported in Table 1 should be easily identified in Fig. 3 (by using a different marker) and what is observed (level or discharge should be mentioned. Fig. 5 is not easy to read Line 199 and line 203. What is meant by "floods were recorded"? Do you mean that a flood gauging was operated instead of using the rating curve?

Authors' answer12 : Figure 3 was modified by using different marker. For line 168 "At Oroftiana gauging station, only the water levels data were registered. And for all other gauging stations are registering, in addition to water level, the discharges data." For line 199 and line 203 Floods were registered at the gauging station.

Referee comment 13: Line 211 the peculiarity of Oancea gauging station and Sivita station distinguishing tidal effects should be documented.

Authors' answer13: At line 211 there is an unfortunate translation for the term "backwaters". "Backwaters" is the correct term instead of "tidal bore". Backwaters were caused by increasing water level of Danube River, which influences the measurements results at the gauging stations situated on the downstream sector of Prut River.

Referee comment 14: Line 243 and elsewhere "Fig. 3 and 6" is not clear. Fig 6 is not easy to read. The peculiarity of Stefanesti(?) station should be mentioned and analyzed in the text. (lines 218 to 221) In all figures, with levels and discherges plts the basin area should be mentioned as well as in lines 310-315.

Authors' answer14 : The figures were modified for a better readability. Stefanesti gaug-

ing station is located in the downstream sector of the dam and itis directly influenced by the discharge water from the Stanca-Costesti Lake (since 1978).

Referee comment 15: Line 316. In is not clear why this mention here "The Oroftiana gauging station only records water level measurements." Idem until line 321. What is the consequence on data accuracy? Line 317. Why this influence?

Authors' answer15 : The water level registered at Radauti Prut gauging station could be influenced by the backwaters caused by Stanca-Costesti Lake. The most obvious case of backwaters was registered during the 2008 historic flood.

Referee comment 16: Lines 329 – 330. Was rainfall observed?

Authors' answer16 : 200-400 mm of rainfall (ie 50-80% of the annual amount) was recorded between 1 May and 15 July 2010. During the flood manifested in 2008, a historic discharge value was registered for Prut river, but the by-passed water volume was low (in upstream of Stanca-Costesti dam) because the flood duration was short. The 2010 flood registered lower maximum discharges compare to 2008, but it by-passed a larger water volume, as flood lasted longer.

Referee comment 17: Line 331-341 should in the study area section

Authors' answer17 : the lines 331-341 were moved in Study area.

Referee comment 18: Line 371; When did this record happened?

Authors' answer18 : (July 5th, 2010)

Referee comment 19: Line 380. Is this increase a result from what was said before?

Authors' answer19 : The discharge increase and the historic values registered were caused by several factors, such as: the water input from the upstream sector of Prut River and the water input added by the Danube backwaters.

Referee comment 20: Line 386 Table 2 should be in the study area section.

Authors' answer20 : This table is better in this paragraph location because the text referee to it.

Referee comment 21: Line 412 the backwater phenomena are effectively very difficult to assess and to predict.

Authors' answer21 : We mentioned this phenomenon because it is unique and had a major local impact for Dorohoi city.

Referee comment 22: Lines 427 to 432. The role of the reservoir and its location in comparison to the river stations is not well described in the text.

Authors' answer22 : The provision of an attenuation water volume (550 million m3) within the lake basin is efficient in retaining a 1% probability flood (reducing it from 2940 m3/s to 700 m3/s). Together with the embankments located on the dam downstream sector, it helps preventing the flooding of 100,000 hectares of meadow. At a normal retention level, Stanca-Costesti lake has a total area of 5900 ha and a water volume of 1.4 billion m3.

Referee comment 23: Line 449 Fig.12 presents challenging issues for water management.

Authors' answer23 : In order to avoid such phenomena it is necessary to increase the height of the overflow structure.

Please also note the supplement to this comment:
http://www.nat-hazards-earth-syst-sci-discuss.net/nhess-2016-289/nhess-2016-289-AC1-supplement.pdf

**Fig. 1.**

**Fig. 2.**

**Fig. 3.**

**Fig. 4.**

Legend

Prut catchment area

*Percent of normal precipitation*
- <50
- 51 - 75
- 76 - 100
- 101 - 125
- 126 - 150
- 151 - 200
- >201

[Figure]

**Fig. 5.**

Fig. 6.

Level (cm): Cernăuți (1), Oroftiana (2), Rădăuți Prut (3), Stânca Aval (4), Ungheni (5), Prisăcani (6), Drânceni (7), Fălciu (8), Oancea (9)

**Legend:**
- ❶ — Cernăuți
- ❷ — Oroftiana
- ❸ — Rădăuți Prut
- ❹ --- Stânca Aval
- ❺ -- Ungheni
- ❻ -- Prisăcani
- ❼ ···· Drânceni
- ❽ ···· Fălciu
- ❾ ···· Oancea

Interactive
comment

**Level (cm)**

500
300
100
0

day | 26 | 27 | 28 | 29 | 30 | 1 | 2 | 3 | 4 | 5 6 7 8 9 10 12 14 16 18 20
month | | | June | | | | | | July

**River, Stream gauges**

| ❶ — Bașeu, Ștefănești | ❹ ––– Milletin, N. Bălcescu | ❼ ····· Bahlui, Holboca |
| ❷ — Buhai, Pădureni | ❺ –– Miletin, Șipote | ❽ ····· Măgura, Cârjoaia |
| ❸ — Sitna, Todireni | ❻ –– Miletin, Hălceni | ❾ ····· Bahlueț, Tg.Frumos |

**Discharge (m³)**

120
80
40
0

day | 26 | 27 | 28 | 29 | 30 | 1 | 2 | 3 | 4 | 5 6 7 8 9 10 12 14 16 18 20
month | | | June | | | | | | July

**Fig. 7.**

[Figure]

[Figure]

[Figure]

[Figure]

**Fig. 8.**

[Figure]

[Figure]

**Fig. 9.**

**Supplement:**

**Exceptional floods in the Prut basin, Romania, in the context of heavy rains in the summer of 2010**

Gheorghe Romanescu[1], Cristian Constantin Stoleriu

Alexandru Ioan Cuza, University of Iasi, Faculty of Geography and Geology, Department of Geography, Bd. Carol I, 20 A, 700505 Iasi, Romania

**Abstract.** The year 2010 was characterized by devastating flooding in Central and Eastern Europe, including Romania, the Czech Republic, Slovakia, and Bosnia-Herzegovina. This study focuses on floods that occurred during the summer of 2010 in the Prut River basin, which has a high percentage of hydrotechnical infrastructure. Strong floods occurred in eastern Romania on the Prut River, which borders the Republic of Moldova and Ukraine, and the Siret River. Atmospheric instability from 21 June-1 July 2010 caused significant amounts of rain, with rates of 51.2 mm/50 min and 42.0 mm/30 min. In the middle Prut basin, there are numerous ponds that help mitigate floods as well as provide water for animals, irrigation, and so forth. The peak discharge of the Prut River during the summer of 2010 was 2,310 m$^3$/s at the Radauti Prut gauging station. High discharges were also recorded on downstream tributaries, including the Baseu, Jijia, and Miletin. High discharges downstream occurred because of water from the middle basin and the backwater from the Danube (a historic discharge of 16,300 m$^3$/s). The floods that occurred in the Prut basin in the summer of 2010 could not be controlled completely because the discharges far exceeded foreseen values.

**1 Introduction**

Catastrophic floods occurred during the summer of 2010 in Central and Eastern Europe. Strong flooding usually occurs at the end of spring and the beginning of summer. Among the most heavily affected countries were Poland, Romania, the Czech Republic, Austria, Germania, Slovakia, Hungary, Ukraine, Serbia, Slovenia, Croatia, Bosnia and Herzegovina, and Montenegro (Bissolli et al., 2011; Szalinska et al., 2014) (Fig. 1). The strongest floods from 2010 were registered in the Danube basin (see Table 1). For Romania, we underlined the floods from the basins of Prut, Siret, Moldova and Bistrita rivers.  The most devastating floods in Romania occurred in Moldavia (Prut, Siret) and Transylvania (Tisa, Somes, Tarnave, Olt). The most deaths were recorded in Poland (25), Romania (six on the Buhai River, a tributary of the Jijia), Slovakia (three), Serbia (two), Hungary (two), and the Czech Republic (two) (Romanescu and Stoleriu, 2013a,b).

Floods are one of the most important natural hazards in Europe (Thieken et al., 2016) and on earth as well (Merz et al., 2010; Riegger et al., 2009). They generate major losses in human lives, and also property damage (Wijkman and Timberlake, 1984).  For this reason, they have been subject to intense research, and significant funds have been allocated to mitigating or stopping them. According to Merz et al. (2010) "the European Flood Directive on the assessment and management of flood risks (European Commission, 2007) requires developing

[1] Corresponding author: romanescugheorghe@gmail.com

management plans for areas with significant flood risk (at a river basin scale), focusing on the reduction of the probability of flooding and on the potential consequences to human health, the environment and economic activity." (p. 511)."This shift in flood risk reduction policies can be observed in the European Flood Directive on the assessment and management of flood risks (European Commission, 2007). It requires developing management plans for areas with significant flood risk, focusing on the reduction of the probability of flooding and of the potential consequences to human health, the environment and economic activity. Flood risk management plans will be integrated in the long term with the river basin management plans of the Water Framework Directive, contributing to integrated water management on the scale of river catchments." (Merz et al., 2010). Several studies investigated catastrophic floods or the floods that generated significant damage. They focused on: the statistical distribution of the maximum annual discharge, using GEV and the links with the basin geology (Ahilan et al., 2012); climate change impacts on floods (Alfieri et al., 2015; Detrembleurs et al., 2015; Schneider et al., 2013; Whitfield, 2012); disastruous effects on infrastructures such as transportation infrastructures, and their interdependence (Berariu et al., 2015); historical floods (Blöschl et al., 2013; Strupczewski et al., 2014; Vasileski and Radevski, 2014) and their links to heavy rainfall (Bostan et al., 2009; Diakakis, 2011; Prudhomme and Genevier, 2011; Retsö, 2015); the public perception of flood risks (Brilly and Polic, 2005; Feldman et al., 2016; Rufat et al., 2015); land use changes and flooding (Cammerer et al., 2012); the evolution of natural risks (Hufschmidt et al., 2005); geomorphological effects of floods in riverbeds (Lichter and Klein, 2011; Lóczy and Gyenizse, 2011; Lóczy et al., 2009, 2014; Reza Ghanbarpour et al., 2014); the spatial distribution of floods (Moel et al., 2009; Parker and Fordham, 1996); the interrelation between snow and flooding (Revuelto et al., 2013).Some of the most interesting studies have investigated catastrophic floods or floods that caused significant damage: statistical distribution of maximum annual discharge using GEV and relationships with basin geology (Ahilan et al., 2012); climate change impacts on floods (Alfieri et al., 2015; Detrembleurs et al., 2015; Schneider et al., 2013; Whitfield, 2012); effects of disasters on infrastructures such as transportation infrastructures and their interdependence (Berariu et al., 2015); historical floods (Blöschl et al., 2013; Strupczewski et al., 2014; Vasileski and Radevski, 2014); relații între precipitații torențiale și inundații istorice (Bostan et al., 2009; Diakakis, 2011; Prudhomme and Genevier, 2011; Retsö, 2015); public perception of flood risks (Brilly and Polic, 2005; Feldman et al., 2016; Rufat et al., 2015); schimbări în utilizarea terenurilor și producerea inundațiilor (Cammerer et al., 2012); evolution of natural risk (Hufschmidt et al., 2005); efecte geomorfologice de albie (Lichter and Klein, 2011; Lóczy and Gyenizse, 2011; Lóczy et al., 2009, 2014; Reza Ghanbarpour et al., 2014); distribuția spațială a inundațiilor (Moel et al., 2009; Parker and Fordham, 1996); interdependența dintre stratul de zăpadă și inundații (Revuelto et al., 2013).

[revised manuscript text omitted]

favoured the formation of a convergent area of humidity. In this case, a layer of humid, warm and unstable air was installed between the topographic surface and 2500 m of altitude. The high quantity of humidity originitated from The Black Sea, situated 500 km away. The warm tropical air is generated by the Russian Plain, overheated by a strong continentality climate. The cold air from medium troposphere, inducted by the cut-off nucleum that generated atmospheric instability, overlapped this structure of the low troposphere (Hustiu, 2011). The synoptic context was disturbed by local physical-geographical factors, especially by the orography of Eastern Carpathians, which led to extremely powerful heavy rains: e.g. 100-200 mm in 24 hours at the sources of Jijia (representing the amount that normally falls during June and July) or 40-60 mm in 24 hours at the Romanian frontier with Ukraine and the Republic of Moldova. The quantity of rainfall in 24 hours were 2-3 higher than the normal values for this period (Hustiu, 2011) (Fig. 4). Majoritatea inundațiilor din România sunt influențate de condițiile climatice care se manifestă la nivel european dar și la nivel local (Birsan, 2015; Birsan and Dumitrescu, 2014; Birsan et al., 2012; Chendes et al., 2015; Corduneanu et al., 2016). În ultima decadă a lunii iunie (20 iunie 2010) și sfârșitul lunii iulie (30 iulie 2010) s-a instalat o zonă baroclină în nordul Moldovei. Aceasta a asigurat formarea unei arii convergente de umezeală. În acest caz între suprafața topografică și altitudinea de 2500m s-a instalat un strat de aer umed, cald și instabil. Cantitatea ridicată de umezeală provine din Marea Neagră, situată la 500 km distanță. Aerul cald tropical este generat de Câmpia Rusă, supraîncălzită ca urmare a continentalismului accentuat. Pe această structură a troposferei joase s-a suprapus aerul rece din troposfera medie, antrenat de nucleul cut-off care a dat naștere instabilității atmosferice (Hustiu, 2011). Contextul sinoptic a fost perturbat de factorii fizico-geografici locali, mai ales de orografia Carpaților Orientali, care au dus la formarea unor ploi torențiale extrem de puternice: 100-200 mm/24 ore la izvoarele râului Jijia (cantitate care cade în mod normal în două luni: iunie și iulie) sau de 40-60 mm/24 ore la frontiera României cu Ucraina și Republica Moldova. Cantitățile de precipitații căzute în 24 de ore depășesc de 2-3 ori normele climatice ale perioadei (Hustiu, 2011) (Fig. ?).

[revised manuscript text omitted]

---

## Author Comment (AC2) · 1 Dec 2016

Dear referee, thank you for your interests about our article,

General comments

The paper copes with the exceptional floods that hit Central Europe and particularly Romania in summer 2010. The work shows interesting flood data for the examined area (though partially presented by the authors in previous works), but it does not constitute a clear contribution to the understanding of these phenomena in the Prut basin, also for its complicated river network. In fact, though the work contains a lot of information on water levels and discharges observed during huge floods, these are mainly

ranked values, roughly compared to similar past events but not statistically defined. In other terms, the paper is too much focussed to the simple inventory of flood values in several gauge stations, and poor attempts to link them to physical reasons or to probabilistic interpretation have been made by authors. Thus, the readability of the paper is not good enough, mainly in the paragraph of the results.

Specific comments

Specifically, though the paper is mainly devoted to flood events, the context of heavy rains of the summer of 2010 (as in the title) is poorly described and could be largely improved. This could be done, for example, by coupling flood diagrams with rainfall histograms, when possible, or by comparing cumulative rainfall values recorded in this event with rainfall that caused other historical floods (also cited in the work). Anyway, the main drawback of the paper is the weak connection between rainfall and floods. In fact, though the period claimed as characterized by intense rainfall is 21 June -1 July 2010, a long set of summer flood (or water level) values is offered to the reader, neither providing any kind of link with triggering precipitation, nor any estimation of the return periods of the rainfall or flood values. Actually, the results are only described by means of simple ranks among critical events. To improve the paper, the paragraph devoted to the results should present at least some evaluations on the estimated frequencies (and not only on critical cases) of the flood values, thus providing more statistical sound to the work. On the other side, the interesting information on water stages and floods overcoming the specific thresholds is described too simply. The valuable data base can be better employed, for example, by combining the temporal overcoming of the higher thresholds in the flood diagrams with the occurrence of the main damages and casualties. This could also provide material for a further interesting discussion on false and missing alarms in the Prut River. Moreover, the work suffers from too much citations, not everywhere appropriate, and from figures affected by some inaccuracies. In brief, though well documented as regards the discharge values, the structure of the work is disorganised enough, with a scarce employment of statistical methodologies

and a long section devoted to the results, which consist principally in a list of flood values, with no link to occurrence frequencies. As a result, the scientific approach of the work is not statistically accurate. Thus, a substantial revision of the paper is needed to improve the quality of the work and provide effectiveness to the flood analysis of the 2010 event in the Prut River.

Technical corrections

Line 18: avoid the word "etc." in the abstract; line 34: change "Earth" with "earth". Lines 61, 153 (and others): I don't understand if the authors use properly the terms "tidal bore" in rivers, except in the case of backwaters actually induced by reservoirs or confluences. Try to be more accurate. Line 76, Figure 2, legend: change "Km" with "km"; avoid decimal ciphers in elevation values. Line 83: it's not clear why the mean altitude assume different values. Line 84: from the figure, the maximum width of Prut basin seems not to be 30 km (even in the lower reaches). Improve the sentence. Lines 101-103: the sentence is trivial (except, maybe, for the presence of the several ponds, which should be recalled). Anyway, the differences among the discharges for the various sections seem very small for such a large river. Lines 107-118: The cited methodologies are not useful for analysing floods, but for recording and collecting data. The paragraph contains too much references and not all perfectly focussed on the issue. The sentence needs a better explanation. Lines 126: it's not usual the call to the Berg intensity scale. If possible, add a reference. Line 132: the CA, CI and CP flood threshold levels should be clearly defined. Line 141: change "1915" with "1914", as noted in the table 1. Line 144, Table 1: the parameter "0 mira level", and mainly its unit "mrBS", should be better explained (or changed). Line 165: the use of the term "significant" should be associated to statistical analysis. Line 175, Figure 4: the values in the legend should not show decimal ciphers. Line 175, Figure 4: can the areal extension of the rainfall analysis be enlarged to the whole Prut basin? Line 177, Figure 5: it's useless to span the graphs before and after the period 20 June – 31 July, that could be better centered with no temporal amplification. Line 235, figure

6: it's not useful to extend the graphs after 1 July. Line 274, figure 7: the temporal amplification can be easily avoided. Line 303, figure 8: the temporal amplification is useless. The legend ("X scale, 0-24 hours") has no meaning. Line 324: the term "significantly" should be associated to statistical analysis. Lines 325-327: the sentence "This value was recalculated." should be better explained. Line 339: the sentence ".allowing the mitigation of 1%." is not clear. Line 373: there are some words repeated ("was eliminated gradually"). Line 430: it can be used directly the acronym "NRL", previously defined in line 335. References in Romanian language should report the words "(in romanian)" at the end of the citation.

Authors' answer: The paper represents an analysis of the situation caused by the flood in 2010 within Prut basin. In the future we intend to analyse the hydrological context of the last 50 years for Prut basin. In this moment we do not have all hydrological data (such as levels, flow rates) for the entire Prut basin. We are in discussion with an official institution in order to obtain hydrological data. The strongest floods from 2010 were registered in the Danube basin (see Table 1). For Romania, we underlined the floods from the basins of Prut, Siret, Moldova and Bistrita rivers. The majority of floods in Romania are influenced by climate factors, which manifest at local and European level (Birsan, 2015; Birsan and Dumitrescu, 2014; Birsan et al., 2012; Chendes et al., 2015; Corduneanu et al., 2016). During the last decade of June (June 20, 2010) and the end of July (July 30, 2010), a baroclinic area was localized in Northern Moldavia. This favored the formation of a convergent area of humidity. In this case, a layer of humid, warm and instable air was installed between the topographic surface and 2500 m of altitude. The high quantity of humidity has its origins from The Black Sea, situated 500 km away. The warm tropical air is generated by the Russian Plain, overheated by a strong continentality climate. The cold air from medium troposphere, inducted by the cut-off nucleus that generated atmospheric instability, overlapped this structure

of the low troposphere (Hustiu, 2011). The synoptic context was disturbed by local physical-geographical factors, especially by the orography of Eastern Carpathians, which led to extremely powerful heavy rains: e.g. 100-200 mm in 24 hours at the sources of Jijia (representing the amount that normally falls during June and July) or 40-60 mm in 24 hours at the Romanian frontier with Ukraine and the Republic of Moldova. The quantity of rainfall during 24 hours were 2-3 higher than the normal values for this period (Hustiu, 2011) (see Figure 4 - Deviation of monthly rainfall amounts (May-July 2010) from the yearly values - CPC, source data NOAA) There were 6 main periods extremely rainy in Romania, located especially in the Moldavian hydrological basins (Prut and Siret): 21-23 June, 25-26 June, 28-30 June, 3-4 July, 6-7 July and 9 July. Rainfall quantities recorded in June were higher. The flash floods registered in Northern Moldavia in 28-29 June 2010 were generated by convective systems with slow spreading. Even if the rainfalls from June 29th were lower, the floods had devastating effects because they came on the context of the increasing water levels from 28 June 2010. Climate convection was organized as a mesocyclone extended over Northern Moldavia (the departments of Suceava and Botosani) (Hustiu, 2011). Methodology: Data on the deviation of rainfall quantities were obtained from the Climate Prediction Center NOOA and from the scientific literature (Hustiu, 2011). Line 18: word "etc." was deleted (Abstract section); Line 34: it was replaced the letter E with e. Lines 61, 153 (and others): it was replaced "tidal bore" with "backwaters". Line 76, "Km" was replaced with "km"; and the decimals from legend were deleted. Line 83: The situation observed at line 83 is an unfortunate manner of writing for describing the mean altitude within Prut catchment basin. The phrase was adjusted as follow: "The mean altitude of the midstream sector of catchment area is 130 m, and for the downstream sector is 2 m.". Line 84: In BrateÈŹ Lake sector is registered 12 km width. Lines 101-103: It's about the water discharge from affluent basins. In this case, the water volumes were cumulated from all the accumulations that contributed to diminishing floods. Lines: 107-118: The paragraph was modified according to the requests of R1. Line 126: Berg et al., 2009. Line 132: These were

explained as requested by R1. Line 141: Changed "1915" with "1914". Line 144, Table 1: "0 nivel mira" was translated to 0 meter level of tide pole and "mrBS stand for meters level reported at Black Sea" Line 165: the term "significantly" was replaced with "high discharge value". Line 175: The decimals from Figure_4's legend were deleted. After de correction operated on article's text and figures, Figure_4 become Figure_5. Line 175, Figure_4 (now Figure_5) represent a zoom on north-eastern part of Romania, where a large amount of precipitations were registered. Lines 177, 235, 274, 303: Figures 5-8 (after de correction operated on article's text and figures, Figures 5-8 become Figures 6-9). Line 324: the term "significantly" was replaced with term "remarkable" Lines 325-327: this value was recalculated through reconstitute discharges. Line 339: The phrase "The reservoir was constructed with a mitigation level of 550 million.m3, allowing the mitigation of a 1% tidal bore from 2,940 to 700 m3/s. The damming infrastructure constructed downstream from the hydrotechnical nodes prevents the flooding of approximately 100,000 ha of floodplain area" was replaced with "The provision of an attenuation water volume (550 million m3) within the lake basin is efficient in retaining a 1% probability flood (reducing it from 2,940 m3/s to 700 m3/s). Together with the embankments located on the dam downstream sector, it helps preventing the flooding of 100,000 hectares of meadow." Line 373: The repeated words were deleted. Line 430: was used directly the acronym "NRL". References in Romanian language were specified with "(in romanian)" at the end of the citation.

Please also note the supplement to this comment:
http://www.nat-hazards-earth-syst-sci-discuss.net/nhess-2016-289/nhess-2016-289-AC2-supplement.pdf

**Fig. 1.**

[Figure]

[Figure]

**Fig. 2.**

**Fig. 3.**

**Fig. 4.**

**Legend**

Prut catchment area

*Percent of normal precipitation*
<50
51 - 75
76 - 100
101 - 125
126 - 150
151 - 200
>201

[Figure]

[Figure]

**Fig. 5.**

[Figure]

Fig. 6.

[Figure]

**Level (cm)**

500
300
100
0
day
month

26 27 28 29 30 1 2 3 4 5 6 7 8 9 10 12 14 16 18 20

June

July

*River, Stream gauges*

| ❶ | Bașeu, Ștefănești | ❹ | Milletin, N. Bălcescu | ❼ | Bahlui, Holboca |
| ❷ | Buhai, Pădureni | ❺ | Miletin, Șipote | ❽ | Măgura, Cârjoaia |
| ❸ | Sitna, Todireni | ❻ | Miletin, Hălceni | ❾ | Bahlueț, Tg.Frumos |

**Discharge (m³)**

120
80
40
0
day
month

26 27 28 29 30 1 2 3 4 5 6 7 8 9 10 12 14 16 18 20

June

July

**Fig. 7.**

[Figure]

[Figure]

[Figure]

**Fig. 8.**

[Figure]

**Fig. 9.**

**Supplement:**

**Exceptional floods in the Prut basin, Romania, in the context of heavy rains in the summer of 2010**

Gheorghe Romanescu[1], Cristian Constantin Stoleriu

Alexandru Ioan Cuza, University of Iasi, Faculty of Geography and Geology, Department of Geography, Bd. Carol I, 20 A, 700505 Iasi, Romania

**Abstract.** The year 2010 was characterized by devastating flooding in Central and Eastern Europe, including Romania, the Czech Republic, Slovakia, and Bosnia-Herzegovina. This study focuses on floods that occurred during the summer of 2010 in the Prut River basin, which has a high percentage of hydrotechnical infrastructure. Strong floods occurred in eastern Romania on the Prut River, which borders the Republic of Moldova and Ukraine, and the Siret River. Atmospheric instability from 21 June-1 July 2010 caused significant amounts of rain, with rates of 51.2 mm/50 min and 42.0 mm/30 min. In the middle Prut basin, there are numerous ponds that help mitigate floods as well as provide water for animals, irrigation, and so forth. The peak discharge of the Prut River during the summer of 2010 was 2,310 m$^3$/s at the Radauti Prut gauging station. High discharges were also recorded on downstream tributaries, including the Baseu, Jijia, and Miletin. High discharges downstream occurred because of water from the middle basin and the backwater from the Danube (a historic discharge of 16,300 m$^3$/s). The floods that occurred in the Prut basin in the summer of 2010 could not be controlled completely because the discharges far exceeded foreseen values.

**1 Introduction**

Catastrophic floods occurred during the summer of 2010 in Central and Eastern Europe. Strong flooding usually occurs at the end of spring and the beginning of summer. Among the most heavily affected countries were Poland, Romania, the Czech Republic, Austria, Germania, Slovakia, Hungary, Ukraine, Serbia, Slovenia, Croatia, Bosnia and Herzegovina, and Montenegro (Bissolli et al., 2011; Szalinska et al., 2014) (Fig. 1). The strongest floods from 2010 were registered in the Danube basin (see Table 1). For Romania, we underlined the floods from the basins of Prut, Siret, Moldova and Bistrita rivers.  The most devastating floods in Romania occurred in Moldavia (Prut, Siret) and Transylvania (Tisa, Somes, Tarnave, Olt). The most deaths were recorded in Poland (25), Romania (six on the Buhai River, a tributary of the Jijia), Slovakia (three), Serbia (two), Hungary (two), and the Czech Republic (two) (Romanescu and Stoleriu, 2013a,b).

Floods are one of the most important natural hazards in Europe (Thieken et al., 2016) and on earth as well (Merz et al., 2010; Riegger et al., 2009). They generate major losses in human lives, and also property damage (Wijkman and Timberlake, 1984).  For this reason, they have been subject to intense research, and significant funds have been allocated to mitigating or stopping them. According to Merz et al. (2010) "the European Flood Directive on the assessment and management of flood risks (European Commission, 2007) requires developing

[1] Corresponding author: romanescugheorghe@gmail.com

management plans for areas with significant flood risk (at a river basin scale), focusing on the reduction of the probability of flooding and on the potential consequences to human health, the environment and economic activity." (p. 511)."This shift in flood risk reduction policies can be observed in the European Flood Directive on the assessment and management of flood risks (European Commission, 2007). It requires developing management plans for areas with significant flood risk, focusing on the reduction of the probability of flooding and of the potential consequences to human health, the environment and economic activity. Flood risk management plans will be integrated in the long term with the river basin management plans of the Water Framework Directive, contributing to integrated water management on the scale of river catchments." (Merz et al., 2010). Several studies investigated catastrophic floods or the floods that generated significant damage. They focused on: the statistical distribution of the maximum annual discharge, using GEV and the links with the basin geology (Ahilan et al., 2012); climate change impacts on floods (Alfieri et al., 2015; Detrembleurs et al., 2015; Schneider et al., 2013; Whitfield, 2012); disastruous effects on infrastructures such as transportation infrastructures, and their interdependence (Berariu et al., 2015); historical floods (Blöschl et al., 2013; Strupczewski et al., 2014; Vasileski and Radevski, 2014) and their links to heavy rainfall (Bostan et al., 2009; Diakakis, 2011; Prudhomme and Genevier, 2011; Retsö, 2015); the public perception of flood risks (Brilly and Polic, 2005; Feldman et al., 2016; Rufat et al., 2015); land use changes and flooding (Cammerer et al., 2012); the evolution of natural risks (Hufschmidt et al., 2005); geomorphological effects of floods in riverbeds (Lichter and Klein, 2011; Lóczy and Gyenizse, 2011; Lóczy et al., 2009, 2014; Reza Ghanbarpour et al., 2014); the spatial distribution of floods (Moel et al., 2009; Parker and Fordham, 1996); the interrelation between snow and flooding (Revuelto et al., 2013).Some of the most interesting studies have investigated catastrophic floods or floods that caused significant damage: statistical distribution of maximum annual discharge using GEV and relationships with basin geology (Ahilan et al., 2012); climate change impacts on floods (Alfieri et al., 2015; Detrembleurs et al., 2015; Schneider et al., 2013; Whitfield, 2012); effects of disasters on infrastructures such as transportation infrastructures and their interdependence (Berariu et al., 2015); historical floods (Blöschl et al., 2013; Strupczewski et al., 2014; Vasileski and Radevski, 2014); relații între precipitații torențiale și inundații istorice (Bostan et al., 2009; Diakakis, 2011; Prudhomme and Genevier, 2011; Retsö, 2015); public perception of flood risks (Brilly and Polic, 2005; Feldman et al., 2016; Rufat et al., 2015); schimbări în utilizarea terenurilor și producerea inundațiilor (Cammerer et al., 2012); evolution of natural risk (Hufschmidt et al., 2005); efecte geomorfologice de albie (Lichter and Klein, 2011; Lóczy and Gyenizse, 2011; Lóczy et al., 2009, 2014; Reza Ghanbarpour et al., 2014); distribuția spațială a inundațiilor (Moel et al., 2009; Parker and Fordham, 1996); interdependența dintre stratul de zăpadă și inundații (Revuelto et al., 2013).

[revised manuscript text omitted]

favoured the formation of a convergent area of humidity. In this case, a layer of humid, warm and unstable air was installed between the topographic surface and 2500 m of altitude. The high quantity of humidity originitated from The Black Sea, situated 500 km away. The warm tropical air is generated by the Russian Plain, overheated by a strong continentality climate. The cold air from medium troposphere, inducted by the cut-off nucleum that generated atmospheric instability, overlapped this structure of the low troposphere (Hustiu, 2011). The synoptic context was disturbed by local physical-geographical factors, especially by the orography of Eastern Carpathians, which led to extremely powerful heavy rains: e.g. 100-200 mm in 24 hours at the sources of Jijia (representing the amount that normally falls during June and July) or 40-60 mm in 24 hours at the Romanian frontier with Ukraine and the Republic of Moldova. The quantity of rainfall in 24 hours were 2-3 higher than the normal values for this period (Hustiu, 2011) (Fig. 4). Majoritatea inundațiilor din România sunt influențate de condițiile climatice care se manifestă la nivel european dar și la nivel local (Birsan, 2015; Birsan and Dumitrescu, 2014; Birsan et al., 2012; Chendes et al., 2015; Corduneanu et al., 2016). În ultima decadă a lunii iunie (20 iunie 2010) și sfârșitul lunii iulie (30 iulie 2010) s-a instalat o zonă baroclină în nordul Moldovei. Aceasta a asigurat formarea unei arii convergente de umezeală. În acest caz între suprafața topografică și altitudinea de 2500m s-a instalat un strat de aer umed, cald și instabil. Cantitatea ridicată de umezeală provine din Marea Neagră, situată la 500 km distanță. Aerul cald tropical este generat de Câmpia Rusă, supraîncălzită ca urmare a continentalismului accentuat. Pe această structură a troposferei joase s-a suprapus aerul rece din troposfera medie, antrenat de nucleul cut-off care a dat naștere instabilității atmosferice (Hustiu, 2011). Contextul sinoptic a fost perturbat de factorii fizico-geografici locali, mai ales de orografia Carpaților Orientali, care au dus la formarea unor ploi torențiale extrem de puternice: 100-200 mm/24 ore la izvoarele râului Jijia (cantitate care cade în mod normal în două luni: iunie și iulie) sau de 40-60 mm/24 ore la frontiera României cu Ucraina și Republica Moldova. Cantitățile de precipitații căzute în 24 de ore depășesc de 2-3 ori normele climatice ale perioadei (Hustiu, 2011) (Fig. ?).

[revised manuscript text omitted]

---

## Author Response (AR1)

**Dear referee, thank you for your interests about our article,**

**Referee#1 comment 1:** Line 34 "Floods are one of the most important natural hazards on Earth" references are about Europe and not the earth

**Authors' answer 1:** Concerning line 34 we omitted to detail the phrase from "Floods are one of the most important natural hazards on Earth" to " Floods are one of the most important natural hazards in Europe (Thieken et al., 2016) and on earth as well (Merz et al., 2010; Riegger et al., 2009). They generate major human life losses and property damage (Wijkman and Timberlake, 1984).", and we modified in text's paper.

**Referee#1 comment 2:** Line 36. "Significant funds...". You may cite the date provided in Merz et al. http://www.nat-hazards-earth-syst-sci.net/nhess- special_issue77-preface.pdf

**Authors' answer2 :** We summarized the ideas specified by Merz et all. Into next paragraph:
"According to Merz et al. (2010) "the European Flood Directive on the assessment and management of flood risks (European Commission, 2007) requires developing management plans for areas with significant flood risk (at a river basin scale), focusing on the reduction of the probability of flooding and of the potential consequences to human health, the environment and economic activity." (p. 511)."

**Referee#1 comment 3:** The reference in lines 37 to 44 should be documented and separated in different topics. Effectively the list is too long and is a mixing of several subjects. For example: -Ahilan et al. 2012 is about statistical distribution of maximum annual discharge using GEV and relationships with basin geology - Alfieri et al. 2015 is about climate change impacts on floods - Berariu et al. 2015 is about the effects of disasters on infrastructures such as transportation infrastructures and their interdependence, etc...

**Authors' answer3:** We rephrase the paragraph about references between lines 37-44
Several studies investigated catastrophic floods or the floods that generated significant damage. They focused on: the statistical distribution of maximum annual discharge, using GEV and the links with the basin geology (Ahilan et al., 2012); climate change impacts on floods (Alfieri et al., 2015; Detrembleurs et al., 2015; Schneider et al., 2013; Whitfield, 2012); disasters effects on infrastructures such as transportation infrastructures, and their interdependence (Berariu et al., 2015); historical floods (Blöschl et al., 2013; Strupczewski et al., 2014; Vasileski and Radevski, 2014) and their links to heavy rain (Bostan et al., 2009; Diakakis, 2011; Prudhomme and Genevier, 2011; Retsö, 2015); public perceptions of flood risks (Brilly and Polic, 2005; Feldman et al., 2016; Rufat et al., 2015); land use changes and flooding (Cammerer et al., 2012); the evolution of natural risks (Hufschmidt et al., 2005);  geomorphological effects of floods in riverbeds (Lichter and Klein, 2011; Lóczy and Gyenizse, 2011; Lóczy et al., 2009, 2014; Reza Ghanbarpour et al., 2014); the spatial distribution of floods (Moel et al., 2009; Parker and Fordham, 1996); the interrelation between snow and flooding (Revuelto et al., 2013).

**Referee#1 comment 4:** Line 61: are the Stanca-Costesti reservoir and the Prut reported in Fig. 1?

**Authors' answer4:** We modified the Figure 1, and also in Fifure 2, in order to appear River Prut, Danube and Stanca-Costesti reservoir.

[Figure]

**Referee#1 comment 5:** Line 83 altitude in the catchment

**Authors' answer5 :** The situation observed at line 83 is an unfortunate manner of writing for describing the mean altitude within Prut catchment basin.  The phrase was adjusted as follow: "The mean altitude of the midstream sector of catchment area is 130 m, and for the downstream sector is 2 m."

**Referee#1 comment 6:** Line 90 Jijia basin area is not documented while this basin is important in the last part of the paper.

**Authors' answer6 :** We introduced some detailed information concerning Jijia River:
"Jijia River has 275 km in length, a catchment area of 5757 km$^2$ and an annual average flow of 14 m3/s. Its most important tributaries are Miletin, Sitna and Bahlui."

**Referee#1 comment 7:** Line 94 what is the criteria to define a "large pond"?

**Authors' answer7 :**
Small ponds are used as drinking water for livestock or to irrigate subsistence rural households. They usually belong to individual households. Large ponds on the other hand have multiple uses, such as: flooding mitigation, irrigation, fish farming etc. They resisted better in time because of their significant surfaces and depths. These large ponds belong to rural or urban communities.

**Referee#1 comment 8:** Line 111 "measurements were taken to estimate the discharge." It is important to say which kind of measurements.

**Authors' answer8:**
Mathematical methods were used to reconstitute discharges and terrain measurements using land surveying equipment (Leica Total Station) were used to calculate the surface of the stream cross-section.

**Referee#1 comment 9:** Lines 113 to 118 Same remark as in lines 37 to 44.  It should be clear what type of method is behind a given reference.  For example Ali et al. (2012) used tracers while Delli-Priscoli and Stakhiv examined "the performance of existing flood protection systems". Line 132 did CA, CI, CP have been defined before?

**Authors' answer9 :** We restructured the paragraph such as:
"The recording and analysing methodology used is standard or slightly adapted to local conditions: e.g. the influence of physical-geographical parameters on runoff (Ali et al., 2012; Kappes et al., 2012; Kourgialas et al., 2012; Waylen and Laporte, 1999); the management of risk situations (Delli-Priscoli and Stakhiv, 2015; Demeritt et al., 2013; Grobicki et al, 2015 Grobicki et al, 2015); the role of reservoirs in flood mitigating (Fu et al., 2014; Serban et al., 2004; Sorocovschi, 2011); the probability of flooding and the changes in the runoff regime (Hall et al., 2004, 2014; Jones, 2011; Seidu et al., 2012a,b; Wu et al., 2011); flood prevention (Hapuarachchi et al., 2011); runoff and streamflow indices (Nguimalet and Ndjendole, 2008); morphologic changes of riverbeds or lake basins (Rusnák and Lehotsky, 2014; Touchart et al., 2012; Verdu et al., 2014) etc."

**Referee#1 comment 10:** Line 148, 149 the methodology should be more detailed.

**Authors' answer10 :**
The cartographic basis used to map altitudes and slopes is Shuttle Radar Topography Mission (Global Land Cover Facility, 2016), at a 1:50000 scale. The vector layers were projected within a geodatabase, using ArcGis 10.1. They include stream lines, sub-catchment basins, and reservoirs and ponds polygons, as well as gauging station points. In order to generate the GIS layers, we applied the following methods: digitisation, queries, conversion, geometries calculation (length, surface) and spatial modelling. Water levels and discharges data were processed and plotted on charts using the Open Office software. We also used the Inkscape software to design the final maps and images.

**Referee#1 comment 11:** Line 154 and 164 are not compatible (1 July, 9 July).

**Authors' answer11 :** In the first case it's about rainfalls registered in Romania (on July 1st) and in the second case it's about those registered in Ukraine (on July 9th).

**Referee#1 comment 12:** Line 168 You need to specify what is registered in each station. What do you mean by "only water levels"? the stations reported in Table 1 should be easily identified in Fig. 3 (by using a different marker) and what is observed (level or discharge should be mentioned. Fig. 5 is not easy to read Line 199 and line 203. What is meant by "floods were recorded"? Do you mean that a flood gauging was operated instead of using the rating curve?

**Authors' answer12 :**
Figure 3 was modified by using different marker.

[Figure]

**For line 168**
"At Oroftiana gauging station, only the water levels data were registered. And for all other gauging stations are registering, in addition to water level, the discharges data."
**For line 199 and line 203**
Floods were registered at the gauging station.

**Referee#1 comment 13:** Line 211 the peculiarity of Oancea gauging station and Sivita station distinguishing tidal effects should be documented.

**Authors' answer13:**
At line 211 there is an unfortunate translation for the term "backwaters". "Backwaters" is the correct term instead of "tidal bore". Backwaters were caused by increasing water level of Danube River, which influences the measurements results at the gauging stations situated on the downstream sector of Prut River.

**Referee#1 comment 14:** Line 243 and elsewhere "Fig. 3 and 6" is not clear. Fig 6 is not easy to read. The peculiarity of Stefanesti(?) station should be mentioned and analyzed in the text. (lines 218 to 221) In all figures, with levels and discherges plts the basin area should be mentioned as well as in lines 310-315.

[Figure]

**Authors' answer14 :**
The figures were modified for a better readability.
Stefanesti gauging station is located in the downstream sector of the dam and itis directly influenced by the discharge water from

the Stanca-Costesti Lake (since 1978).

**Referee#1 comment 15:** Line 316. In is not clear why this mention here "The Oroftiana gauging station only records water level measurements." Idem until line 321. What is the consequence on data accuracy? Line 317. Why this influence?

**Authors' answer15 :** The water level registered at Radauti Prut gauging station could be influenced by the backwaters caused by Stanca-Costesti Lake. The most obvious case of backwaters was registered during the 2008 historic flood.

**Referee#1 comment 16:** Lines 329 – 330. Was rainfall observed?

**Authors' answer 16 :**
200-400 mm of rainfall (ie 50-80% of the annual amount) was recorded between 1 May and 15 July 2010. During the flood manifested in 2008, a historic discharge value was registered for Prut river, but the by-passed water volume was low (in upstream of Stanca-Costesti dam) because the flood duration was short. The 2010 flood registered lower maximum discharges compare to 2008, but it by-passed a larger water volume, as flood lasted longer.

**Referee#1 comment 17:** Line 331-341 should in the study area section

**Authors' answer 17 :** the lines 331-341 were moved in Study area.

**Referee#1 comment 18:** Line 371; When did this record happened?

**Authors' answer 18 :** (July 5th, 2010)

**Referee#1 comment 19:** Line 380. Is this increase a result from what was said before?

**Authors' answer 19 :**
The discharge increase and the historic values registered were caused by several factors, such as: the water input from the upstream sector of Prut River and the water input added by the Danube backwaters.

**Referee#1 comment 20:** Line 386 Table 2 should be in the study area section.

**Authors' answer20 :** This table is better in this paragraph location because the text referee to it.

**Referee#1 comment 21:** Line 412 the backwater phenomena are effectively very difficult to assess and to predict.

**Authors' answer21 :** We mentioned this phenomenon because it is unique and had a major local impact for Dorohoi city.

**Referee#1 comment 22:** Lines 427 to 432. The role of the reservoir and its location in comparison to the river stations is not well described in the text.

**Authors' answer22 :**
The provision of an attenuation water volume (550 million $m^3$) within the lake basin is efficient in retaining a 1% probability flood (reducing it from 2940 $m^3/s$ to 700 $m^3/s$). Together with the embankments located on the dam downstream sector, it helps preventing the flooding of 100,000 hectares of meadow. At a normal retention level, Stanca-Costesti lake has a total area of 5900 ha and a water volume of 1.4 billion $m^3$.

**Referee#1 comment 23:** Line 449 Fig.12 presents challenging issues for water management.

**Authors' answer23 :**
In order to avoid such phenomena it is necessary to increase the height of the overflow structure.

Nat. Hazards Earth Syst. Sci. Discuss., doi:10.5194/nhess-2016-289-RC2, 2016 © Author(s) 2016. CC-BY 3.0 License.
The paper copes with the exceptional floods that hit Central Europe and particularly Romania in summer 2010. The work shows interesting flood data for the examined area (though partially presented by the authors in previous works), but it does not constitute a clear contribution to the understanding of these phenomena in the Prut basin, also for its complicated river network. In fact, though the work contains a lot of information on water levels and discharges observed during huge floods, these are mainly ranked values, roughly compared to similar past events but not statistically defined. In other terms, the paper is too much focussed to the simple inventory of flood values in several gauge stations, and poor attempts to link them to physical reasons or to probabilistic interpretation have been made by authors. Thus, the readability of the paper is not good enough, mainly in the paragraph of the results.

Specific comments

Specifically, though the paper is mainly devoted to flood events, the context of heavy rains of the summer of 2010 (as in the title) is poorly described and could be largely improved. This could be done, for example, by coupling flood diagrams with rainfall histograms, when possible, or by comparing cumulative rainfall values recorded in this event with rainfall that caused other historical floods (also cited in the work). Anyway, the main drawback of the paper is the weak connection between rainfall and floods. In fact, though the period claimed as characterized by intense rainfall is 21 June -1 July 2010, a long set of summer flood (or water level) values is offered to the reader, neither providing any kind of link with triggering precipitation, nor any estimation of the return periods of the rainfall or flood values. Actually, the results are only described by means of simple ranks among critical events. To improve the paper, the paragraph devoted to the results should present at least some evaluations on the estimated frequencies (and not only on critical cases) of the flood values, thus providing more statistical sound to the work. On the other side, the interesting information on water stages and floods overcoming the specific thresholds is described too simply. The valuable data base can be better employed, for example, by combining the temporal overcoming of the higher thresholds in the flood diagrams with the occurrence of the main damages and casualties. This could also provide material for a further interesting discussion on false and missing alarms in the Prut River. Moreover, the work suffers from too much citations, not everywhere appropriate, and from figures affected by some inaccuracies. In brief, though well documented as regards the discharge values, the structure of the work is disorganised enough, with a scarce employment of statistical methodologies and a long section devoted to the results, which consist principally in a list of flood values, with no link to occurrence frequencies. As a result, the scientific approach of the work is not statistically accurate. Thus, a substantial revision of the paper is needed to improve the quality of the work and provide effectiveness to the flood analysis of the 2010 event in the Prut River.

**Technical corrections**

Line 18: avoid the word "etc." in the abstract; line 34: change "Earth" with "earth". Lines 61, 153 (and others): I don't understand if the authors use properly the terms "tidal bore" in rivers, except in the case of backwaters actually induced by reservoirs or confluences. Try to be more accurate. Line 76, Figure 2, legend: change "Km" with "km"; avoid decimal ciphers in elevation values. Line 83: it's not clear why the mean altitude assume different values. Line 84: from the figure, the maximum width of Prut basin seems not to be 30 km (even in the lower reaches). Improve the sentence. Lines 101-103: the sentence is trivial (except, maybe, for the presence of the several ponds, which should be recalled). Anyway, the differences among the discharges for the various sections seem very small for such a large river. Lines 107-118: The cited methodologies are not useful for analysing floods, but for recording and collecting data. The paragraph contains too much references and not all perfectly focussed on the issue. The sentence needs a better explanation. Lines 126: it's not usual the call to the Berg intensity scale. If possible, add a reference. Line 132: the CA, CI and CP flood threshold levels should be clearly defined. Line 141: change "1915" with "1914", as noted in the table 1. Line 144, Table 1: the parameter "0 mira level", and mainly its unit "mrBS", should be better explained (or changed). Line 165: the use of the term "significant" should be associated to statistical analysis. Line 175, Figure 4: the values in the legend should not show decimal ciphers. Line 175, Figure 4: can the areal extension of the rainfall analysis be enlarged to the whole Prut basin? Line 177, Figure 5: it's useless to span the graphs before and after the period 20 June – 31 July, that could be better centered with no temporal amplification. Line 235, figure 6: it's not useful to extend the graphs after 1 July. Line 274, figure 7: the temporal amplification can be easily avoided. Line 303, figure 8: the temporal amplification is useless. The legend ("X scale, 0-24 hours") has no meaning. Line 324: the term "significantly" should be associated to statistical analysis. Lines 325-327: the sentence "This value was recalculated." should be better explained. Line 339: the sentence ". allowing the mitigation of 1%." is not clear. Line 373: there are some words repeated ("was eliminated gradually"). Line 430: it can be used directly the acronym "NRL", previously defined in line 335. References in Romanian language should report the words "(in romanian)" at the end of the citation.

**Authors' answer:**
The paper represents an analysis of the situation caused by the flood in 2010 within Prut basin. In the future we intend to analyse the hydrological context of the last 50 years for Prut basin. In this moment we do not have all hydrological data (such as levels, flow rates) for the entire Prut basin. We are in discussion with an official institution in order to obtain hydrological data.

The strongest floods from 2010 were registered in the Danube basin (see Table 1). For Romania, we underlined the floods from the basins of Prut, Siret, Moldova and Bistrita rivers.

The majority of floods in Romania are influenced by climate factors, which manifest at local and European level (Birsan, 2015; Birsan and Dumitrescu, 2014; Birsan et al., 2012; Chendes et al., 2015; Corduneanu et al., 2016). During the last decade of June (June 20, 2010) and the end of July (July 30, 2010), a baroclinic area was localized in Northern Moldavia. This favored the formation of a convergent area of humidity. In this case, a layer of humid, warm and instable air was installed between the topographic surface and 2500 m of altitude. The high quantity of humidity has its origins from The Black Sea, situated 500 km away. The warm tropical air is generated by the Russian Plain, overheated by a strong continentality climate. The cold air from medium troposphere, inducted by the cut-off nucleum that generated atmospheric instability, overlapped this structure of the low troposphere (Hustiu, 2011). The synoptic context was disturbed by local physical-geographical factors, especially by the orography of Eastern Carpathians, which led to extremely powerful heavy rains: e.g. 100-200 mm in 24 hours at the sources of Jijia (representing the amount that normally falls during June and July) or 40-60 mm in 24 hours at the Romanian frontier with Ukraine and the Republic of Moldova. The quantity of rainfall during 24 hours were 2-3 higher than the normal values for this period (Hustiu, 2011) (see Figure 4).

[Figure]

Deviation of monthly rainfall amounts (May-July 2010) from the yearly values - CPC (source data NOAA)

There were 6 main periods extremely rainy in Romania, located especially in the Moldavian hydrological basins (Prut and Siret): 21-23 June, 25-26 June, 28-30 June, 3-4 July, 6-7 July and 9 July. Rainfall quantities recorded in June were higher. The flash floods registered in Northern Moldavia in 28-29 June 2010 were generated by convective systems with slow spreading. Even if the rainfalls from June 29th were lower, the floods had devastating effects because they came on the context of the increasing water levels from 28 June 2010. Climate convection was organized as a mesocyclone extended over Northern Moldavia (the departments of Suceava and Botosani) (Hustiu, 2011).

**Methodology:** Data on the deviation of rainfall quantities were obtained from the Climate Prediction Center NOOA and from the scientific literature (Hustiu, 2011).

**Line 18:** word "etc." was deleted (Abstract section);

**Line 34:** it was replaced the letter E with e.

**Lines 61, 153** (and others): it was replaced "tidal bore" with "backwaters".

**Line 76**, "Km" was replaced with "km"; and the decimals from legend were deleted.

**Line 83:**

The situation observed at line 83 is an unfortunate manner of writing for describing the mean altitude within Prut catchment basin. The phrase was adjusted as follow: "The mean altitude of the midstream sector of catchment area is 130 m, and for the downstream sector is 2 m.".

**Line 84:** In Brateş Lake sector is registered 12 km width.

**Lines 101-103:** It's about the water discharge from affluent basins. In this case, the water volumes were cumulated from all the

accumulations that contributed to diminishing floods.

**Lines: 107-118:** The paragraph was modified according to the requests of R1.

**Line 126:** Berg et al., 2009.

**Line 132:** These were explained as requested by R1.

**Line 141:** Changed "1915" with "1914".

**Line 144**, Table 1: "0 nivel mira" was translated to 0 meter level of tide pole and "mrBS stand for meters level reported at Black Sea"

**Line 165:** the term "significantly" was replaced with "high discharge value".

**Line 175**: The decimals from Figure_4's legend were deleted. After de correction operated on article's text and figures, Figure_4 become Figure_5.

[Figure]

**Line 175**, Figure_4 (now Figure_5) represent a zoom on north-eastern part of Romania, where a large amount of precipitations were registered.

**Lines 177, 235, 274, 303:** Figures 5-8 (after de correction operated on article's text and figures, Figures 5-8 become Figures 6-9).

**Line 324:** the term "significantly" was replaced with term "remarkable"

[Figure]

**Lines 325-327**: this value was recalculated through reconstitute discharges.

Line 339: The phrase "The reservoir was constructed with a mitigation level of 550 million.m³, allowing the mitigation of a 1% tidal bore from 2,940 to 700 m³/s. The damming infrastructure constructed downstream from the hydrotechnical nodes prevents the flooding of approximately 100,000 ha of floodplain area" was replaced with "The provision of an attenuation water volume (550

million m3) within the lake basin is efficient in retaining a 1% probability flood (reducing it from 2,940 m3/s to 700 m3/s). Together with the embankments located on the dam downstream sector, it helps preventing the flooding of 100,000 hectares of meadow."

**Line 373:** The repeated words were deleted.

**Line 430:** was used directly the acronym "NRL".

References in Romanian language were specified with "(in romanian)" at the end of the citation.

**Exceptional floods in the Prut basin, Romania, in the context of heavy rains in the summer of 2010**

Gheorghe Romanescu[1], Cristian Constantin Stoleriu

Alexandru Ioan Cuza, University of Iasi, Faculty of Geography and Geology, Department of Geography, Bd. Carol I, 20 A, 700505 Iasi, Romania

**Abstract.** The year 2010 was characterized by devastating flooding in Central and Eastern Europe, including Romania, the Czech Republic, Slovakia, and Bosnia-Herzegovina. This study focuses on floods that occurred during the summer of 2010 in the Prut River basin, which has a high percentage of hydrotechnical infrastructure. Strong floods occurred in eastern Romania on the Prut River, which borders the Republic of Moldova and Ukraine, and the Siret River. Atmospheric instability from 21 June-1 July 2010 caused remarkable amounts of rain, with rates of 51.2 mm/50 min and 42.0 mm/30 min. In the middle Prut basin, there are numerous ponds that help mitigate floods as well as provide water for animals, irrigation, and so forth. The peak discharge of the Prut River during the summer of 2010 was 2,310 $m^3$/s at the Radauti Prut gauging station. High discharges were also recorded on downstream tributaries, including the Baseu, Jijia, and Miletin . High discharges downstream occurred because of water from the middle basin and the backwater from the Danube (a historic discharge of 16,300 $m^3$/s). The floods that occurred in the Prut basin in the summer of 2010 could not be controlled completely because the discharges far exceeded foreseen values.

**1 Introduction**

Catastrophic floods occurred during the summer of 2010 in Central and Eastern Europe. Strong flooding usually occurs at the end of spring and the beginning of summer. Among the most heavily affected countries were Poland, Romania, the Czech Republic, Austria, Germania, Slovakia, Hungary, Ukraine, Serbia, Slovenia, Croatia, Bosnia and Herzegovina, and Montenegro (Bissolli et al., 2011; Szalinska et al., 2014) (Fig. 1). The strongest floods from 2010 were registered in the Danube basin (see Table 1). For Romania, we underlined the floods from the basins of Prut, Siret, Moldova and Bistrita rivers.  The most devastating floods in Romania occurred in Moldavia (Prut, Siret) and Transylvania (Tisa, Somes, Tarnave, Olt). The most deaths were recorded in Poland (25), Romania (six on the Buhai River, a tributary of the Jijia), Slovakia (three), Serbia (two), Hungary (two), and the Czech Republic (two) (Romanescu and Stoleriu, 2013a,b).

Floods are one of the most important natural hazards in Europe (Thieken et al., 2016) and on earth as well (Merz et al., 2010; Riegger et al., 2009). They generate major losses in human lives, and also property damage (Wijkman and Timberlake, 1984).  For this reason, they have been subject to intense research, and significant funds have been allocated to mitigating or stopping them. According to Merz et al. (2010) "the European Flood Directive on the
* * *
[1] Corresponding author: romanescugheorghe@gmail.com

assessment and management of flood risks (European Commission, 2007) requires developing management plans for areas with significant flood risk (at a river basin scale), focusing on the reduction of the probability of flooding and on the potential consequences to human health, the environment and economic activity." (p. 511). This shift in flood risk reduction policies can be observed in the European Flood Directive on the assessment and management of flood risks (European Commission, 2007). It requires developing management plans for areas with significant flood risk, focusing on the reduction of the probability of flooding and of the potential consequences to human health, the environment and economic activity. Flood risk management plans will be integrated in the long term with the river basin management plans of the Water Framework Directive, contributing to integrated water management on the scale of river catchments." (Merz et al., 2010). Several studies investigated catastrophic floods or the floods that generated significant damage. They focused on: the statistical distribution of the maximum annual discharge, using GEV and the links with the basin geology (Ahilan et al., 2012); climate change impacts on floods (Alfieri et al., 2015; Detrembleurs et al., 2015; Schneider et al., 2013; Whitfield, 2012); disastruous effects on infrastructures such as transportation infrastructures, and their interdependence (Berariu et al., 2015); historical floods (Blöschl et al., 2013; Strupczewski et al., 2014; Vasileski and Radevski, 2014) and their links to heavy rainfall (Bostan et al., 2009; Diakakis, 2011; Prudhomme and Genevier, 2011; Retsö, 2015); the public perception of flood risks (Brilly and Polic, 2005; Feldman et al., 2016; Rufat et al., 2015); land use changes and flooding (Cammerer et al., 2012); the evolution of natural risks (Hufschmidt et al., 2005); geomorphological effects of floods in riverbeds (Lichter and Klein, 2011; Lóczy and Gyenizse, 2011; Lóczy et al., 2009, 2014; Reza Ghanbarpour et al., 2014); the spatial distribution of floods (Moel et al., 2009; Parker and Fordham, 1996); the interrelation between snow and flooding (Revuelto et al., 2013). Some of the most interesting studies have investigated catastrophic floods or floods that caused significant damage: statistical distribution of maximum annual discharge using GEV and relationships with basin geology (Ahilan et al., 2012); climate change impacts on floods (Alfieri et al., 2015; Detrembleurs et al., 2015; Schneider et al., 2013; Whitfield, 2012); effects of disasters on infrastructures such as transportation infrastructures and their interdependence (Berariu et al., 2015); historical floods (Blöschl et al., 2013; Strupczewski et al., 2014; Vasileski and Radevski, 2014); relații între precipitații torențiale și inundații istorice (Bostan et al., 2009; Diakakis, 2011; Prudhomme and Genevier, 2011; Retsö, 2015); public perception of flood risks (Brilly and Polic, 2005; Feldman et al., 2016; Rufat et al., 2015); schimbări în utilizarea terenurilor și producerea inundațiilor (Cammerer et al., 2012); evolution of natural risk (Hufschmidt et al., 2005); efecte geomorfologice de albie (Lichter and Klein, 2011; Lóczy and Gyenizse, 2011; Lóczy et al., 2009, 2014; Reza Ghanbarpour et al., 2014); distribuția spațială a inundațiilor (Moel et al., 2009; Parker and Fordham, 1996); interdependența dintre stratul de zăpadă și inundații (Revuelto et al., 2013).

[revised manuscript text omitted]

High discharge and water levels of 2,310 m³/s and 744 cm (+144 cm CP), respectively, were recorded at the Radauti Prut gauging station. The 2010 values are remarkablesignificantly lower than the maximum values recorded in 2008 of 7,140 m³/s and 1,130 cm (+530 cm CP) (the highest value for Romanian rivers). This value was recalculated after two years (through recomposed discharges)(prin intermediul debitelor reconstituite), resulting in a discharge of 4,240 m³/s, which is the second highest value in Romania (after the historic discharge of 4,650 m³/s on the Siret in 2005) (Romanescu et al., 2011a,b). The existence of five backwatertidal bore peaks (with the second and third backwatertidal bores being weaker) clearly indicates that they were caused by heavy rains in the Carpathian Mountains in Ukraine. A volume of 200-400 mm of rainfall (ie 50-80% of the annual amount) was recorded between 1 May and 15 July 2010. During the flood manifested in 2008, a historic discharge value was registered for Prut River, but the by-passed water volume was low (in upstream of Stanca-Costesti dam) because the flood duration was short. The 2010 flood registered lower maximum discharges compare to 2008, but it by-passed a larger water volume, as flood lasted longer.În perioada 1 mai 15 iulie 2010 s au înregistrat precipitații cuprinse între 200 400 mm (adică 50 80% din norma anuală). Viitura din anul 2008 a înregistrat debitul istoric pentru râul Prut dar volumul de apă tranzitat a fost redus (amonte de barajul Stânca Costești) deoarece durata fenomenului a fost scurtă. Viitura din anul 2010 a înregistrat debite maxime mai reduse dar a tranzitat un volum mai mare de apă deoarece durata fenomenului a fost îndelungată.

[revised manuscript text omitted]
$^3$ Prevederea unui volum de apă de atenuare (550 milioane m$^3$) în cadrul lacului face ca viitura cu probabilitate de 1% să fie atenuată de la 2940 m$^3$/s la 700 m$^3$/s. Împreună cu îndiguirile efectuate în aval de baraj se evită inundarea a 100000 ha de luncă. La Nivelul Normal de retenție lacul însumează o suprafață de 5900 ha și un volum de apă de 1400 milioane m$^3$.

Discharges downstream of the Stanca-Costesti reservoir are controlled by reservoirs and retention systems constructed on the main tributaries of the Prut. We emphasize that the Jijia and Bahlui catchments have hydrotechnical works on 80% of their surface areas. The system of polders in the downstream sector of the Jijia River was used extensively to mitigate discharge and prevent the city of Galati from flooding (Galati is the largest Danubian port, situated at the confluence of the Prut and the Danube Rivers).

The gauging stations in the lower sector of the Prut recorded high discharges and water levels because of excess water coming from upstream (the middle sector of the Prut). At the Oancea gauging station, however, which is situated near the discharge of the Prut into the Danube, there is a significant backwater influence. The Danube had historic discharge at Galati, which affected the water level at Oancea station on the Prut.

Floods during the summer of 2010, in northeast Romania, rank third among hydrological disasters in Romanian history after the floods of 2005 and 2008, which also occurred in the Siret and Prut catchments. The 2010 floods caused grave economic damage (almost one billion Euros in just the Prut catchment) and greatly affected agriculture. Furthermore, six people died in Dorohoi, on the Buhai River.

**Figure 12.** The "spider flow" phenomenon in which the Buhai waters climbed the Ezer dam on the Jijia, in the area of confluence of the two rivers

The 2010 floods caused a unique backwater phenomenon at the mouth of the Buhai River. Floodwaters from the Buhai climbed the Ezer dam (situated on the Jijia River) and flooded its lacustrine cuvette. The phenomenon was called "spider flow". In order to avoid such phenomena it is necessary to increase the height of the overflow structure.The phenomenon was called "spider flow". Pentru evitarea unor asemenea fenomene este necesară supraînălțarea deversorului de ape mari.

[revised manuscript text omitted]

Font color: Black, English (U.K.)

| Page 4: [2] Formatted | Admin | 12/1/2016 2:01:00 AM |
|---|---|---|

Font: 10 pt, Bold, Font color: Black, English (U.K.)

| Page 4: [3] Formatted | Admin | 11/28/2016 7:23:00 PM |
|---|---|---|

Indent: Left:  0.2 cm, Right:  0.2 cm

| Page 4: [4] Formatted | Admin | 12/1/2016 2:01:00 AM |
|---|---|---|

Font: Bold, Font color: Black

| Page 4: [5] Formatted | Admin | 12/1/2016 2:01:00 AM |
|---|---|---|

Font: 10 pt, Bold, Font color: Black, English (U.K.)

| Page 4: [6] Formatted | Admin | 12/1/2016 2:01:00 AM |
|---|---|---|

Font: Bold, Font color: Black

| Page 4: [7] Formatted | Admin | 12/1/2016 2:01:00 AM |
|---|---|---|

Font: 10 pt, Bold, Font color: Black, English (U.K.)

| Page 4: [8] Formatted | Admin | 12/1/2016 2:01:00 AM |
|---|---|---|

Font: Bold, Font color: Black

| Page 4: [9] Formatted | Admin | 12/1/2016 2:01:00 AM |
|---|---|---|

Font: 10 pt, Bold, Font color: Black, English (U.K.)

| Page 4: [10] Formatted | Admin | 12/1/2016 2:01:00 AM |
|---|---|---|

Font: 10 pt, Font color: Black, English (U.K.)

| Page 4: [11] Formatted Table | Admin | 11/28/2016 7:26:00 PM |
|---|---|---|

Formatted Table

| Page 4: [12] Formatted | Admin | 12/1/2016 2:01:00 AM |
|---|---|---|

Font color: Black

| Page 4: [12] Formatted | Admin | 12/1/2016 2:01:00 AM |
|---|---|---|

Font color: Black

| Page 4: [13] Formatted | Admin | 12/1/2016 2:01:00 AM |
|---|---|---|

Font color: Black

| Page 4: [13] Formatted | Admin | 12/1/2016 2:01:00 AM |
|---|---|---|

Font color: Black

| Page 4: [13] Formatted | Admin | 12/1/2016 2:01:00 AM |
|---|---|---|

Font color: Black

| Page 4: [13] Formatted | Admin | 12/1/2016 2:01:00 AM |
|---|---|---|

Font color: Black

| Page 4: [14] Formatted | Admin | 12/1/2016 2:01:00 AM |
|---|---|---|

Font color: Black

| Page 4: [14] Formatted | Admin | 12/1/2016 2:01:00 AM |
|---|---|---|

Font color: Black

| Page 4: [15] Formatted | Admin | 12/1/2016 2:01:00 AM |
|---|---|---|

Font color: Black

| Page 4: [15] Formatted | Admin | 12/1/2016 2:01:00 AM |
|---|---|---|

Font color: Black

| Page 4: [15] Formatted | Admin | 12/1/2016 2:01:00 AM |
|---|---|---|

Font color: Black

| Page 4: [15] Formatted | Admin | 12/1/2016 2:01:00 AM |
|---|---|---|

Font color: Black

| Page 4: [16] Formatted | Admin | 11/28/2016 7:26:00 PM |
|---|---|---|

Space After:  0 pt

| Page 4: [17] Formatted | Admin | 12/1/2016 2:01:00 AM |
|---|---|---|

Font: 10 pt, Font color: Black, English (U.K.)

| Page 4: [17] Formatted | Admin | 12/1/2016 2:01:00 AM |
|---|---|---|

Font: 10 pt, Font color: Black, English (U.K.)

| Page 4: [17] Formatted | Admin | 12/1/2016 2:01:00 AM |
|---|---|---|

Font: 10 pt, Font color: Black, English (U.K.)

| Page 4: [18] Formatted | Admin | 12/1/2016 2:01:00 AM |
|---|---|---|

Font color: Black

| Page 4: [18] Formatted | Admin | 12/1/2016 2:01:00 AM |
|---|---|---|

Font color: Black

| Page 4: [19] Formatted | Admin | 12/1/2016 2:01:00 AM |
|---|---|---|

Font color: Black

| Page 4: [19] Formatted | Admin | 12/1/2016 2:01:00 AM |
|---|---|---|

Font color: Black

| Page 4: [20] Formatted | Admin | 12/1/2016 2:01:00 AM |
|---|---|---|

Font: 10 pt, Font color: Black, English (U.K.)

| Page 4: [20] Formatted | Admin | 12/1/2016 2:01:00 AM |
|---|---|---|

Font: 10 pt, Font color: Black, English (U.K.)

| Page 4: [20] Formatted | Admin | 12/1/2016 2:01:00 AM |
|---|---|---|

Font: 10 pt, Font color: Black, English (U.K.)

| Page 4: [21] Formatted | Admin | 12/1/2016 2:01:00 AM |
|---|---|---|

Font color: Black

| Page 4: [21] Formatted | Admin | 12/1/2016 2:01:00 AM |
|---|---|---|

Font color: Black

| Page 4: [22] Formatted | Admin | 12/1/2016 2:01:00 AM |
|---|---|---|

Font color: Black

| Page 4: [22] Formatted | Admin | 12/1/2016 2:01:00 AM |
|---|---|---|

Font color: Black

| Page 4: [23] Formatted | Admin | 12/1/2016 2:01:00 AM |
|---|---|---|

Font color: Black

| Page 4: [23] Formatted | Admin | 12/1/2016 2:01:00 AM |
|---|---|---|

Font color: Black

| Page 4: [24] Formatted | Admin | 12/1/2016 2:01:00 AM |
|---|---|---|

Font: 10 pt, Font color: Black, English (U.K.)

| Page 4: [24] Formatted | Admin | 12/1/2016 2:01:00 AM |
|---|---|---|

Font: 10 pt, Font color: Black, English (U.K.)

| Page 4: [24] Formatted | Admin | 12/1/2016 2:01:00 AM |
|---|---|---|

Font: 10 pt, Font color: Black, English (U.K.)

| Page 4: [25] Formatted | Admin | 12/1/2016 2:01:00 AM |
|---|---|---|

Font color: Black

| Page 4: [25] Formatted | Admin | 12/1/2016 2:01:00 AM |
|---|---|---|

Font color: Black

| Page 4: [26] Formatted | Admin | 12/1/2016 2:01:00 AM |
|---|---|---|

Font color: Black

| Page 4: [26] Formatted | Admin | 12/1/2016 2:01:00 AM |
|---|---|---|

Font color: Black

| Page 4: [27] Formatted | Admin | 12/1/2016 2:01:00 AM |
|---|---|---|

Font color: Black

| Page 4: [27] Formatted | Admin | 12/1/2016 2:01:00 AM |
|---|---|---|

Font color: Black

| Page 4: [28] Formatted | Admin | 11/28/2016 7:26:00 PM |
|---|---|---|

Space After:  0 pt

| Page 4: [29] Formatted | Admin | 12/1/2016 2:01:00 AM |
|---|---|---|

Font: 10 pt, Font color: Black, English (U.K.)

| Page 4: [29] Formatted | Admin | 12/1/2016 2:01:00 AM |
|---|---|---|

Font: 10 pt, Font color: Black, English (U.K.)

| Page 4: [29] Formatted | Admin | 12/1/2016 2:01:00 AM |
|---|---|---|

Font: 10 pt, Font color: Black, English (U.K.)

| Page 4: [30] Formatted | Admin | 12/1/2016 2:01:00 AM |
|---|---|---|

Font color: Black

| Page 4: [30] Formatted | Admin | 12/1/2016 2:01:00 AM |
|---|---|---|

Font color: Black

| Page 4: [31] Formatted | Admin | 12/1/2016 2:01:00 AM |
|---|---|---|

Font color: Black

| Page 4: [31] Formatted | Admin | 12/1/2016 2:01:00 AM |

Font color: Black

| Page 4: [31] Formatted | Admin | 12/1/2016 2:01:00 AM |

Font color: Black

| Page 4: [31] Formatted | Admin | 12/1/2016 2:01:00 AM |

Font color: Black

| Page 4: [32] Formatted | Admin | 12/1/2016 2:01:00 AM |

Font color: Black

| Page 4: [32] Formatted | Admin | 12/1/2016 2:01:00 AM |

Font color: Black

| Page 4: [33] Formatted | Admin | 12/1/2016 2:01:00 AM |

Font color: Black

| Page 4: [33] Formatted | Admin | 12/1/2016 2:01:00 AM |

Font color: Black

| Page 4: [34] Formatted | Admin | 11/28/2016 7:26:00 PM |

Space After:  0 pt

| Page 4: [35] Formatted | Admin | 12/1/2016 2:01:00 AM |

Font: 10 pt, Font color: Black, English (U.K.)

| Page 4: [35] Formatted | Admin | 12/1/2016 2:01:00 AM |

Font: 10 pt, Font color: Black, English (U.K.)

| Page 4: [35] Formatted | Admin | 12/1/2016 2:01:00 AM |

Font: 10 pt, Font color: Black, English (U.K.)

| Page 4: [36] Formatted | Admin | 12/1/2016 2:01:00 AM |

Font color: Black

| Page 4: [36] Formatted | Admin | 12/1/2016 2:01:00 AM |

Font color: Black

| Page 4: [37] Formatted | Admin | 12/1/2016 2:01:00 AM |

Font color: Black

| Page 4: [37] Formatted | Admin | 12/1/2016 2:01:00 AM |

Font color: Black

| Page 4: [38] Formatted | Admin | 12/1/2016 2:01:00 AM |

Font color: Black

| Page 4: [38] Formatted | Admin | 12/1/2016 2:01:00 AM |

Font color: Black

| Page 4: [38] Formatted | Admin | 12/1/2016 2:01:00 AM |

Font color: Black

| Page 4: [38] Formatted | Admin | 12/1/2016 2:01:00 AM |

Font color: Black

| Page 4: [38] Formatted | Admin | 12/1/2016 2:01:00 AM |

Font color: Black

| Page 4: [38] Formatted | Admin | 12/1/2016 2:01:00 AM |

Font color: Black

| Page 4: [39] Formatted | Admin | 12/1/2016 2:01:00 AM |

Font color: Black

| Page 4: [39] Formatted | Admin | 12/1/2016 2:01:00 AM |

Font color: Black

| Page 4: [40] Formatted | Admin | 11/28/2016 7:26:00 PM |

Space After:  0 pt

| Page 4: [41] Formatted | Admin | 12/1/2016 2:01:00 AM |

Font: 10 pt, Font color: Black, English (U.K.)

| Page 4: [42] Formatted | Admin | 12/1/2016 2:01:00 AM |

Font color: Black

| Page 4: [42] Formatted | Admin | 12/1/2016 2:01:00 AM |

Font color: Black

| Page 4: [42] Formatted | Admin | 12/1/2016 2:01:00 AM |

Font color: Black

| Page 4: [42] Formatted | Admin | 12/1/2016 2:01:00 AM |

Font color: Black

| Page 4: [42] Formatted | Admin | 12/1/2016 2:01:00 AM |

Font color: Black

| Page 4: [42] Formatted | Admin | 12/1/2016 2:01:00 AM |

Font color: Black

| Page 4: [43] Formatted | Admin | 12/1/2016 2:01:00 AM |

Font color: Black

| Page 4: [43] Formatted | Admin | 12/1/2016 2:01:00 AM |

Font color: Black

| Page 4: [44] Formatted | Admin | 11/28/2016 7:26:00 PM |

Space After:  0 pt

| Page 4: [45] Formatted | Admin | 12/1/2016 2:01:00 AM |

Font: 10 pt, Font color: Black, English (U.K.)

| Page 4: [45] Formatted | Admin | 12/1/2016 2:01:00 AM |

Font: 10 pt, Font color: Black, English (U.K.)

| Page 4: [45] Formatted | Admin | 12/1/2016 2:01:00 AM |

Font: 10 pt, Font color: Black, English (U.K.)

| Page 4: [46] Formatted | Admin | 12/1/2016 2:01:00 AM |

Font color: Black

| Page 4: [46] Formatted | Admin | 12/1/2016 2:01:00 AM |

Font color: Black

| Page 4: [47] Formatted | Admin | 12/1/2016 2:01:00 AM |

Font color: Black

| Page 4: [47] Formatted | Admin | 12/1/2016 2:01:00 AM |

Font color: Black

| Page 4: [48] Formatted | Admin | 12/1/2016 2:01:00 AM |

Font color: Black

| Page 4: [48] Formatted | Admin | 12/1/2016 2:01:00 AM |

Font color: Black

| Page 4: [49] Formatted | Admin | 12/1/2016 2:01:00 AM |

Font: 10 pt, Font color: Black, English (U.K.)

| Page 4: [49] Formatted | Admin | 12/1/2016 2:01:00 AM |

Font: 10 pt, Font color: Black, English (U.K.)

| Page 4: [49] Formatted | Admin | 12/1/2016 2:01:00 AM |

Font: 10 pt, Font color: Black, English (U.K.)

| Page 4: [50] Formatted | Admin | 12/1/2016 2:01:00 AM |

Font color: Black

| Page 4: [50] Formatted | Admin | 12/1/2016 2:01:00 AM |

Font color: Black

| Page 4: [51] Formatted | Admin | 12/1/2016 2:01:00 AM |

Font color: Black

| Page 4: [51] Formatted | Admin | 12/1/2016 2:01:00 AM |

Font color: Black

| Page 4: [52] Formatted | Admin | 12/1/2016 2:01:00 AM |

Font color: Black

| Page 4: [52] Formatted | Admin | 12/1/2016 2:01:00 AM |

Font color: Black

| Page 4: [53] Formatted | Admin | 12/1/2016 2:01:00 AM |

Font color: Black

| Page 4: [53] Formatted | Admin | 12/1/2016 2:01:00 AM |

Font color: Black

| Page 4: [54] Formatted | Admin | 12/1/2016 2:01:00 AM |

Font: 10 pt, Font color: Black, English (U.K.)

| Page 4: [54] Formatted | Admin | 12/1/2016 2:01:00 AM |

Font: 10 pt, Font color: Black, English (U.K.)

| Page 4: [54] Formatted | Admin | 12/1/2016 2:01:00 AM |

Font: 10 pt, Font color: Black, English (U.K.)

| Page 4: [55] Formatted | Admin | 12/1/2016 2:01:00 AM |

Font color: Black

| Page 4: [55] Formatted | Admin | 12/1/2016 2:01:00 AM |

Font color: Black

| Page 4: [56] Formatted | Admin | 12/1/2016 2:01:00 AM |

Font color: Black

| Page 4: [56] Formatted | Admin | 12/1/2016 2:01:00 AM |

Font color: Black

| Page 4: [57] Formatted | Admin | 12/1/2016 2:01:00 AM |

Font: 10 pt, Font color: Black, English (U.K.)

| Page 4: [57] Formatted | Admin | 12/1/2016 2:01:00 AM |

Font: 10 pt, Font color: Black, English (U.K.)

| Page 4: [57] Formatted | Admin | 12/1/2016 2:01:00 AM |

Font: 10 pt, Font color: Black, English (U.K.)

| Page 4: [58] Formatted | Admin | 12/1/2016 2:01:00 AM |

Font color: Black

| Page 4: [58] Formatted | Admin | 12/1/2016 2:01:00 AM |

Font color: Black

| Page 4: [59] Formatted | Admin | 12/1/2016 2:01:00 AM |

Font color: Black

| Page 4: [59] Formatted | Admin | 12/1/2016 2:01:00 AM |

Font color: Black

| Page 4: [60] Formatted | Admin | 12/1/2016 2:01:00 AM |

Font: 10 pt, Font color: Black, English (U.K.)

| Page 4: [60] Formatted | Admin | 12/1/2016 2:01:00 AM |

Font: 10 pt, Font color: Black, English (U.K.)

| Page 4: [60] Formatted | Admin | 12/1/2016 2:01:00 AM |

Font: 10 pt, Font color: Black, English (U.K.)

| Page 4: [61] Formatted | Admin | 12/1/2016 2:01:00 AM |

Font color: Black

| Page 4: [61] Formatted | Admin | 12/1/2016 2:01:00 AM |

Font color: Black

| Page 4: [62] Formatted | Admin | 12/1/2016 2:01:00 AM |

Font color: Black

| Page 4: [62] Formatted | Admin | 12/1/2016 2:01:00 AM |

Font color: Black

| Page 4: [63] Formatted | Admin | 12/1/2016 2:01:00 AM |

Font: 10 pt, Font color: Black, English (U.K.)

| Page 4: [63] Formatted | Admin | 12/1/2016 2:01:00 AM |

Font: 10 pt, Font color: Black, English (U.K.)

| Page 4: [63] Formatted | Admin | 12/1/2016 2:01:00 AM |

Font: 10 pt, Font color: Black, English (U.K.)

| Page 4: [64] Formatted | Admin | 12/1/2016 2:01:00 AM |

Font color: Black

| Page 4: [64] Formatted | Admin | 12/1/2016 2:01:00 AM |

Font color: Black

| Page 4: [65] Formatted | Admin | 12/1/2016 2:01:00 AM |

Font color: Black

| Page 4: [65] Formatted | Admin | 12/1/2016 2:01:00 AM |

Font color: Black

| Page 4: [66] Formatted | Admin | 12/1/2016 2:01:00 AM |

Font color: Black

| Page 4: [66] Formatted | Admin | 12/1/2016 2:01:00 AM |

Font color: Black

---

## Author Response (AR2)

Editor Decision: Publish subject to technical corrections (13 Feb 2017) by Dr. Maria-Carmen Llasat

Comments to the Author:

Dear Dr. Romanescu,

It is a pleasure for me telling you that your paper "Exceptional floods in the Prut basin, Romania, in the context of heavy rains in the summer of 2010" by Gheorghe Romanescu, Cristian Constantin Stoleriu, has been accepted subject to technical corrections before its publication. Please take into consideration the comments made by the referee as well as my own comments in the final version of your paper. I include them in the bottom up of my message.

Hoping to receive more contributions from you in the future, to be published in NHESS,

Sincerely,

Prof. Maria Carmen Llasat

Technical modifications

Line 163: "with a mitigation level of 550 million.m3 » change with « with a mitigation volume of 550 million m3"

Author's Answer: "with a mitigation level of 550 million.m3 » was replaced with « with a mitigation volume of 550 million m3"

Lines 174-176: change "Mathematical methods were used to reconstitute discharges and terrain measurements using land surveying equipment (Leica Total Station) were used to calculate the surface of the stream cross-section." By "Mathematical methods were used to reconstitute discharges and terrain measurements using land surveying equipment (Leica Total Station) to calculate the surface of the stream cross-section".

Author's Answer: "Mathematical methods were used to reconstitute discharges and terrain measurements using land surveying equipment (Leica Total Station) were used to calculate the surface of the stream cross-section." was replaced with "Mathematical methods were used to reconstitute discharges and terrain measurements using land surveying equipment (Leica Total Station) to calculate the surface of the stream cross-section".

Line 618: you speak about probability flood of 1% but previously in line 164 it was about backwater.

Author's Answer: In both cases It is about 1% probability flood.

Line 644 A brief mention and description of Ezer dam (and this dam objectives in the water system) should be given in the system's description early in the text.

Author's Answer: A brief description of Ezer dam was introduced in line 109.

Lines 203-205. Please, substitute the present sentences "....and unstable air was installed between the topographic surface and 2500 m of altitude. The high quantity of humidity originitated from The

Black Sea, situated 500 km away. The warm 205 tropical air is generated by the Russian Plain, overheated by a strong continentality climate" by "...and unstable air was installed between the surface and 2500 m of altitude. The high quantity of humidity was originated from the Black Sea, situated 500 km away. The warm air was generated in the Russian Plain, overheated by a strong continentality climate".

Author's Answer: "....and unstable air was installed between the topographic surface and 2500 m of altitude. The high quantity of humidity originitated from The Black Sea, situated 500 km away. The warm 205 tropical air is generated by the Russian Plain, overheated by a strong continentality climate" was replaced with "...and unstable air was installed between the surface and 2500 m of altitude. The high quantity of humidity was originated from the Black Sea, situated 500 km away. The warm air was generated in the Russian Plain, overheated by a strong continentality climate".

In the legend of figure 4 write "Deviation of monthly rainfall amounts (May-July 2010) from the average cumulated precipitation for this period". I understand that the percentage refers to the comparison with average cumulated precipitation for May-June-July. If you refer to annual precipitation, please, indicate it in the Legend of the figure.

Author's Answer: Figure 4. "Deviation of monthly rainfall amounts (May-July 2010)…" was replaced with Figure 4. Cumulative precipitation for May-July (2010) interval, divided by normal precipitation-Climate Prediction Center (source data: NOAA)

Lines 222-224. Please, substitute the present sentence "The climate convection was organized as a mesocyclone extended over Northern Moldavia (the departments of Suceava and Botosani) (Hustiu, 2011)" by "The convection was organized by a mesocyclone extended over Northern Moldavia (the departments of Suceava and Botosani) (Hustiu, 2011)"

Author's Answer: Please, substitute the present sentence "The climate convection was organized as a mesocyclone extended over Northern Moldavia (the departments of Suceava and Botosani) (Hustiu, 2011)" was replaced with "The convection was organized by a mesocyclone extended over Northern Moldavia (the departments of Suceava and Botosani) (Hustiu, 2011)"

Lines 506-507. Please, substitute the present sentence "The greatest damage occurred in, and the most arable area was destroyed in, the middle Prut basin in the Jijia-Bahlui Depression of the Moldavian Plain" by "The greatest damage occurred in the middle Prut basin in the Jijia-Bahlui Depression of the Moldavian Plain, where the most arable area was destroyed".

Author's Answer: Please, substitute the present sentence "The greatest damage occurred in, and the most arable area was destroyed in, the middle Prut basin in the Jijia-Bahlui Depression of the Moldavian Plain" was replaced with "The greatest damage occurred in the middle Prut basin in the Jijia-Bahlui Depression of the Moldavian Plain, where the most arable area was destroyed".